# A specific G9a inhibitor unveils BGLT3 lncRNA as a universal mediator of chemically induced fetal globin gene expression

Shohei Takase[1], Takashi Hiroyama[2], Fumiyuki Shirai[3], Yuki Maemoto[1], Akiko Nakata[4], Mayumi Arata[5], Seiji Matsuoka[4], Takeshi Sonoda[4], Hideaki Niwa[6], Shin Sato[6], Takashi Umehara[6], Mikako Shirouzu[6], Yosuke Nishigaya[7], Tatsunobu Sumiya[7], Noriaki Hashimoto[7], Ryosuke Namie[7], Masaya Usui[8], Tomokazu Ohishi[9], Shun-ichi Ohba[9], Manabu Kawada[9], Yoshihiro Hayashi[10], Hironori Harada[10], Tokio Yamaguchi[11], Yoichi Shinkai[12], Yukio Nakamura[2], Minoru Yoshida[4,5,13]✉ & Akihiro Ito[1,5]✉

Sickle cell disease (SCD) is a heritable disorder caused by β-globin gene mutations. Induction of fetal γ-globin is an established therapeutic strategy. Recently, epigenetic modulators, including G9a inhibitors, have been proposed as therapeutic agents. However, the molecular mechanisms whereby these small molecules reactivate γ-globin remain unclear. Here we report the development of a highly selective and non-genotoxic G9a inhibitor, RK-701. RK-701 treatment induces fetal globin expression both in human erythroid cells and in mice. Using RK-701, we find that BGLT3 long non-coding RNA plays an essential role in γ-globin induction. RK-701 selectively upregulates BGLT3 by inhibiting the recruitment of two major γ-globin repressors in complex with G9a onto the BGLT3 gene locus through CHD4, a component of the NuRD complex. Remarkably, BGLT3 is indispensable for γ-globin induction by not only RK-701 but also hydroxyurea and other inducers. The universal role of BGLT3 in γ-globin induction suggests its importance in SCD treatment.

In jawed vertebrates, the expression of β-like globin genes gradually supersedes the expression of other globins during development, which is known as "globin switching." In humans, switching from embryonic (HbE, α2ε2) to fetal (HbF, α2γ2) globins occurs in utero, and then a transition from fetal to adult (HbA, α2β2) globins occurs after birth. β-hemoglobinopathies, including sickle cell disease (SCD), are autosomal recessive disorders caused by point mutations in the β-globin gene (*HBB*). HbF reactivation in adult erythroid cells has been

[1]Laboratory of Cell Signaling, School of Life Sciences, Tokyo University of Pharmacy and Life Sciences, Hachioji, Tokyo 192-0392, Japan. [2]Cell Engineering Division, RIKEN BioResource Research Center, Tsukuba, Ibaraki 305-0074, Japan. [3]Drug Discovery Chemistry Platform Unit, RIKEN Center for Sustainable Resource Science, Wako, Saitama 351-0198, Japan. [4]Drug Discovery Seed Compounds Exploratory Unit, RIKEN Center for Sustainable Resource Science, Wako, Saitama 351-0198, Japan. [5]Chemical Genomics Research Group, RIKEN Center for Sustainable Resource Science, Wako, Saitama 351-0198, Japan. [6]Drug Discovery Structural Biology Platform Unit, RIKEN Center for Biosystems Dynamics Research, Yokohama, Kanagawa 230-0045, Japan. [7]Watarase Research Center, Discovery Research Headquarters, Kyorin Pharmaceutical Co. Ltd., Shimotsuga-gun, Tochigi 329-0114, Japan. [8]Support Unit for Bio-Material Analysis, Research Resources Division, RIKEN Center for Brain Science, Wako, Saitama 351-0198, Japan. [9]Institute of Microbial Chemistry (BIKAKEN), Numazu, Microbial Chemistry Research Foundation, Numazu, Shizuoka 410-0301, Japan. [10]Laboratory of Oncology, School of Life Sciences, Tokyo University of Pharmacy and Life Sciences, Hachioji, Tokyo 192-0392, Japan. [11]RIKEN Program for Drug Discovery and Medical Technology Platforms, Yokohama, Kanagawa 230-0045, Japan. [12]Cellular Memory Laboratory, Cluster for Pioneering Research, Wako, Saitama 351-0198, Japan. [13]Department of Biotechnology, the University of Tokyo, Bunkyo-ku, Tokyo 113-8657, Japan. ✉e-mail: yoshidam@riken.jp; aito@toyaku.ac.jp

well established as a therapeutic strategy for these diseases[1]. Indeed, hydroxyurea, which has HbF-inducing activity, has been used as an approved drug for SCD, but it has limited efficacy as well as safety problems[1,2]. Therefore, efforts have been made to find new methods to induce HbF[3].

Five β-like globin genes, including embryonic ε (*HBE1*), homologous fetal ^Aγ and ^Gγ (*HBG1* and *HBG2*), adult δ (*HBD*), and *HBB*, are linearly arranged in their order of expression during development. The locus control region (LCR), located 40–60 kb upstream of the β-like globin gene locus, plays an important role in globin switching by changing loop structures and interacting with β-like globin promoters through a heteropentamer complex consisting of GATA1, TAL1, E2A, LMO2, and LDB1[4]. In addition, a number of nuclear regulators, including transcriptional regulators and non-coding RNAs, are suggested to be involved in globin switching[5]. Among them, a genome-wide association study and subsequent functional analyses identified BCL11A as a major negative regulator of γ-globin expression[6]. BCL11A occupies the β-like globin gene locus through specific DNA sequence motifs and functions as a transcriptional repressor. BCL11A silences the γ-globin gene by enhancing long-range interaction between the LCR and the β-globin gene cluster by modulating chromosomal loop formation[7]. In addition to BCL11A, ZBTB7A (also known as LRF) has emerged as another important transcriptional repressor that functions independently of BCL11A[8]. ZBTB7A recognizes DNA sequence motifs that differ from those for BCL11A and that directly suppress γ-globin transcription[9].

In addition to nuclear transcriptional factors, noncoding RNAs, including long noncoding RNAs (lncRNAs), are implicated in globin switching. BGLT3 lncRNA is a known globin-switching regulator[10]. The BGLT3 gene is located in the intergenic region between *HBD* and *HBG1* and is dynamically co-transcribed with γ-globin genes in erythroid cells. BGLT3 lncRNA has been shown to act as a positive regulator of γ-globin transcription by interacting with the mediator complex on chromatin. However, the functional relationship between BGLT3 lncRNA and other factors such as the LCR, BCL11A, and ZBTB7A in the regulation of γ-globin expression is poorly understood. In addition, the suitability of BGLT3 as a target for fetal γ-globin induction therapy remains unclear.

Developmental globin switching is also regulated by epigenetic machineries such as histone modifications and DNA methylation. Silencing of γ-globin is associated with DNA methylation and a decrease in active histone modifications[11,12]. Both BCL11A and ZBTB7A regulate chromatin structure by interacting with the NuRD complex containing histone deacetylase (HDAC) 1 and 2, resulting in the repression of γ-globin transcription[8,13,14]. Upregulation of γ-globin has consistently been observed when epigenetic modulators such as DNA methyltransferases[14–16], HDACs[17,18], histone lysine demethylase LSD1[19], histone arginine methyltransferase PRMT5[20], and histone lysine methyltransferases G9a and G9a-like protein (GLP)[21,22] are genetically or pharmacologically inhibited. G9a and GLP catalyze the dimethylation of histone H3 lysine 9 (H3K9me2), which acts as an epigenetic repressive mark. UNC0638, the most commonly used inhibitor of G9a and GLP[23] has been shown to induce HbF in erythroid cells[21,22]. Thus, G9a has emerged as an attractive target for SCD therapy. However, the use of currently available G9a inhibitors for therapeutic purposes seems limited by their toxicity[21,22]. In addition, the molecular mechanisms by which G9a inhibition induces γ-globin expression are still unclear.

Here, we develop RK-701, which is a small-molecule inhibitor of G9a and GLP with high selectivity and low toxicity. It reactivates fetal globin expression both in vitro and in vivo and reduces H3K9me2 levels. Importantly, the use of RK-701 as a chemical probe to investigate the mechanistic function of G9a/GLP in γ-globin gene regulation demonstrate that BGLT3 lncRNA plays an essential role in the reactivation of fetal globin gene expression in erythroid cells, not only by RK-701 but also by other chemical inducers.

## Results

### Development of RK-701, a specific and low-toxicity G9a inhibitor

To identify G9a inhibitors with better selectivity and safety profiles, we performed high-throughput screening (HTS) using an in vitro fluorogenic assay[24]. This screening identified several small molecules that inhibited the enzymatic activity of G9a (Supplementary Fig. 1). Among the hit compounds, we confirmed that compound 1 inhibits G9a with an IC50 value of 5.3 μM using an amplified luminescent proximity homogeneous assay-linked immunosorbent assay (AlphaLISA)-based assay (Supplementary Table 1)[24]. Structure-based optimization of compound 1 was performed (Supplementary Table 1). Compound 1 has a chiral carbon atom, and its *S*-enantiomer (compound 2) exhibited stronger inhibitory activity than racemic compound 1. The replacement of the thioether group with the bulky benzene group (compound 3) caused a loss of inhibitory activity. Deletion (compound 4) or alteration (compounds 5–7) of the cyclohexanone moiety, or modification of the benzimidazole group (compound 8) dramatically reduced inhibitory activity. On the other hand, the replacement of the benzimidazole group with the aniline amide group (compound 9) increased the inhibitory activity. Further optimization studies finally generated RK-701 (compound 10), which potently inhibited G9a with an IC50 value of 23–27 nM, and GLP with an IC50 value of 53 nM (Fig. 1a, b, and Supplementary Tables 1 and 2). Since a significant number of derivatives were synthesized based on fragment molecular orbital (FMO) calculations and their structure-activity relationships (SAR) were investigated, the details of the optimization process will be reported elsewhere.

Like compound 1, RK-701 has a chiral center, and its *R*-enantiomer, RK-0133114, exhibited ~100-fold weaker inhibition than RK-701 (Fig. 1a and b). Surface plasmon resonance (SPR) analysis confirmed the reversible binding of RK-701 to G9a but failed to detect the binding of RK-133114 to G9a, even at high concentrations (Supplementary Fig. 2). The inhibitory activity of RK-701 against G9a and GLP was over 1,000-fold stronger than against other methyltransferases (Supplementary Table 2). The crystal structure of the G9a–RK-701 complex (Supplementary Table 3, PDB ID: 7X73) elucidated the mode of RK-701 binding to the substrate-binding groove, in which the central cyclopropylethyl branch was inserted into the lysine channel (Fig. 1c, d). The pyrrole nitrogen and the adjacent carbamoyl oxygen and nitrogen atoms form hydrogen bonds with the main chain atoms of Asp1088 and Tyr1154. The pyrrole ring forms a π-π stack with the side chain of Phe1087. In addition, the aniline amide forms a hydrogen bond with the carbonyl oxygen of Leu1086, and its benzene ring forms a T-shaped π-π stack with the side chain of Phe1158. Superimposing the structures of G9a–RK-701 and G9a–UNC0638 (PDB ID: 3RJW)[23] demonstrated that RK-701 has a binding mode that differs from that of UNC0638, even though both inhibitors block the lysine channel (Fig. 1e). Kinetic analyses also support the idea that RK-701 is a histone H3 substrate-competitive inhibitor because the IC50 value increased linearly with elevating the histone H3 peptide concentration (Supplementary Fig. 3a). On the other hand, a decrease in the IC50 value with increasing *S*-adenosyl methionine (SAM) concentration suggests uncompetitive inhibition with respect to SAM (Supplementary Fig. 3b). The reason for the difference between IC50 (27 nM) and Kd (750 nM) values is currently unknown, but its mode of uncompetitive inhibition with SAM may affect these values because IC50 and Kd values were measured in the presence and absence of SAM, respectively[25–28].

Both the parallel artificial membrane permeability assay (PAMPA) and Caco-2 permeability assay revealed a decent cell permeability of RK-701 (Supplementary Table 4). A dose-dependent reduction of cellular H3K9me2 levels was observed in human umbilical cord blood-derived erythroid progenitor 2 (HUDEP-2) cells upon treatment with RK-701 but not RK-0133114 (Fig. 1f), which was consistent with the potency of in vitro enzyme inhibition (Fig. 1b). On the other hand, RK-701 did not affect the expression levels of both G9a and GLP (Fig. 1f). In

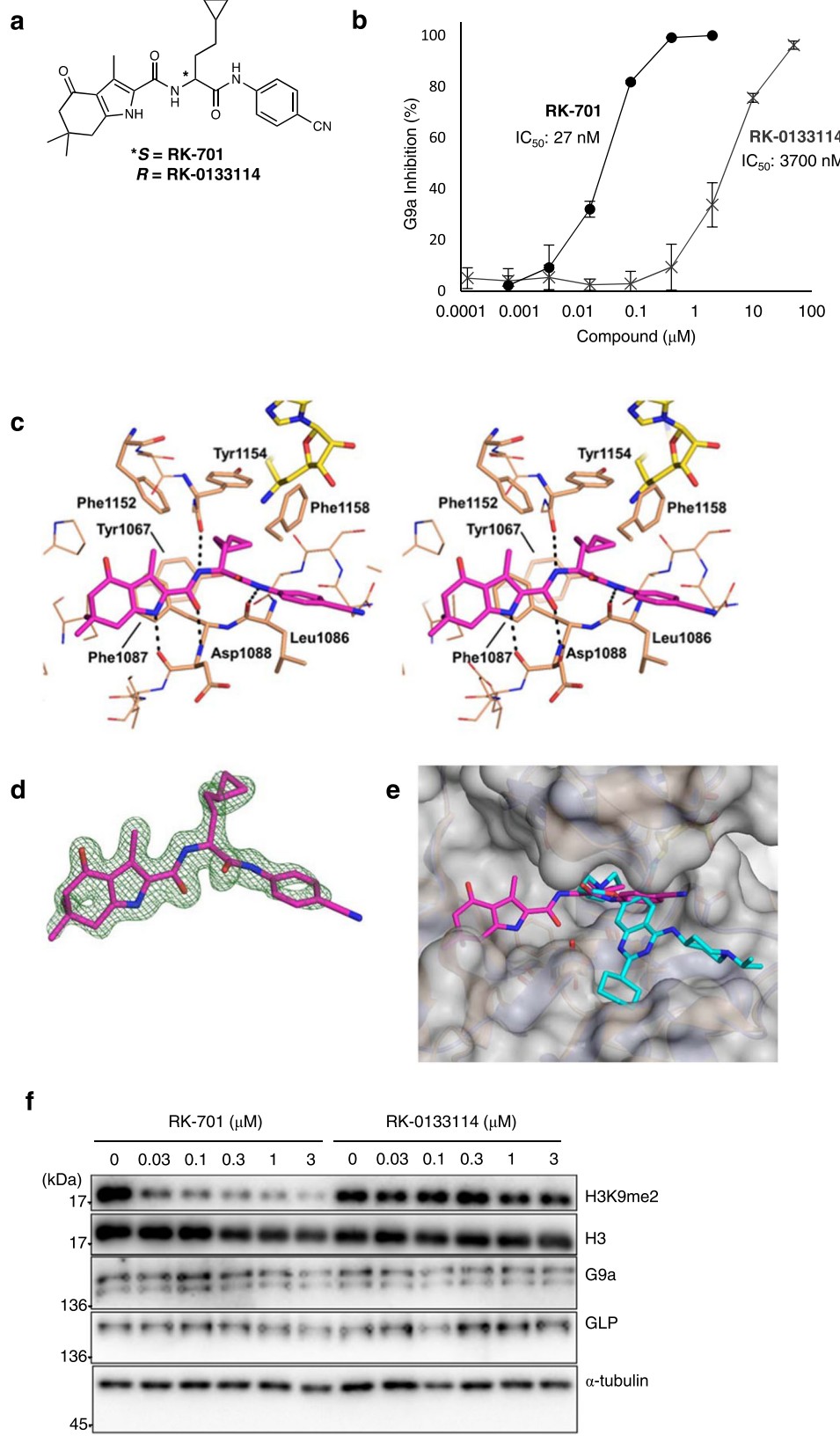

contrast to commercially available G9a inhibitors such as UNC0638, RK-701 showed essentially no cytotoxic effect on the viability of normal cells, including rat myoblast cell line H9c2 and HUDEP-2 cells (Supplementary Fig. 4a and b)[23,29]. In addition, an acute toxicity assay showed that all groups of mice treated with RK-701 survived for two weeks, whereas those treated with all other G9a inhibitors died at high

dose ranges (Supplementary Fig. 4c). Furthermore, no significant body weight loss was observed in mice treated with RK-701 (Supplementary Fig. 4d). We also evaluated potential mutagenic and genotoxic activities of RK-701 using the Ames test and the in vivo micronucleus test, respectively. RK-701 exhibited no mutagenicity in either the TA98 or TA100 strains of *Salmonella typhimurium* regardless of metabolic

**Fig. 1 | Development of a specific G9a inhibitor RK-701. a** Structures of RK-701 (*S*-form) and its enantiomer RK-0133114 (*R*-form). The asterisk indicates the chiral center. **b** Potencies of RK-701 and RK-0133114 (shown in black and gray, respectively) to inhibit G9a activity. Data are mean ± SD from three independent experiments. **c** The binding mode of RK-701 to the G9a substrate-binding site. The complex of G9a and RK-701 is shown in stereoview. RK-701, G9a, and sinefungin molecules are shown in magenta, brown, and yellow, respectively. G9a residues that interact with RK-701 are shown by a stick model, and hydrogen bonds are shown by dotted black lines. **d** An omit mFo−DFc map contoured at 2.5 σ is shown by the green mesh. **e** Superimposition of the G9a−RK-701 (PDB ID: 7X73) and G9a−UNC0638 (PDB ID: 3RJW) structures is shown. RK-701 and UNC0638 molecules are colored in magenta and cyan, respectively. The molecular surface of RK-701−bound G9a is shown in gray. The side chain of Arg1157 located above RK-701 is omitted from the surface calculation to clearly show the binding site. **f** Effects of RK-701 and RK-0133114 on H3K9me2 in HUDEP-2 cells. HUDEP-2 cells were treated with different concentrations of RK-701 or RK-0133114 for 4 days. Lysates were immunoblotted with the indicated antibodies. A representative image of three different analyses is shown.

activation (Supplementary Fig. 4e). Oral administration of RK-701 at up to 100 mg/kg for two weeks did not cause the formation of micronuclei in bone marrow cells of rats (Supplementary Fig. 4f) and had no serious effect on hematological and hepatic parameters in rats (Supplementary Table 5). All these results indicate that RK-701 is a reversible, specific, and cell-permeable G9a inhibitor that has essentially no acute toxicity or genotoxicity under the dose of 100 mg/kg and that is likely much safer than existing G9a inhibitors such as UNC0638.

### Induction of fetal globin by treatment with RK-701

We next examined the effect of RK-701 on γ-globin reactivation in HUDEP-2 cells. RK-701 upregulated the mRNA level of γ-globin, but not β-globin (Fig. 2a), and increased the percentage of HbF-expressing cells (Supplementary Fig. 5a) in a concentration-dependent manner. On the other hand, the inactive *R*-enantiomer RK-0133114 failed to increase HbF in HUDEP-2 cells (Fig. 2a and Supplementary Fig. 5a), suggesting that RK-701 induced HbF by inhibiting cellular G9a/GLP methyltransferase activity. The upregulation of γ-globin expression by RK-701 was further evaluated using primary human CD34[+] hematopoietic cells. RK-701 significantly increased γ-globin mRNA expression (Fig. 2b), the level of HbF expression (Fig. 2c and Supplementary Fig. 5b−f), and the percentage of HbF-expressing cells (Fig. 2d) without affecting cell viability or erythroid differentiation (Supplementary Fig. 5g and h), in a dose range similar to that which decreased the cellular H3K9me2 level (Fig. 2e). On the other hand, hydroxyurea, which did not affect the H3K9me2 level (Fig. 2e), was less effective in inducing HbF than RK-701 (Fig. 2b−d) and showed severe cytotoxicity (Supplementary Fig. 5g). Next, we investigated whether RK-701 exhibited in vivo activity in mice. Mouse embryonic εy-globin, which corresponds to human fetal globin, was selectively increased in RK-701−treated mice and there was a significant decrease in H3K9me2 levels in peripheral blood cells (Fig. 2f, g). Consistent with these observations, the in vivo pharmacokinetic analysis demonstrated that a single intraperitoneal administration (25 mg/kg) of RK-701 resulted in a maximum plasma concentration ($C_{max}$) of 42.4 μM, which was sufficient to inhibit G9a activity (Supplementary Fig. 6). Also, RK-701 was acceptably stable in liver microsomes from various species and most stable in human liver microsomes (Supplementary Table 4). Notably, the five-week continuous administration of RK-701 had no effect on mouse body weights (Fig. 2h), further indicating the low toxicity of this compound. Together, these results suggest that RK-701 is a bioavailable and safe G9a inhibitor with excellent potential as a therapeutic agent against SCD.

### The essential role of BGLT3 in γ-globin reactivation by G9a inhibition

Although the efficacy of RK-701 proved that G9a/GLP inhibition can induce HbF, the molecular mechanism of this process remains unclear. First, we examined the effect of RK-701 on histone H3K9me2 levels at the β-globin gene locus using ChIP-seq analysis with an antibody against H3K9me2. The occupancies of H3K9me2 were reduced by RK-701 throughout the β-globin gene locus (Fig. 3a, b). These results were confirmed by ChIP-quantitative PCR (qPCR) analyses (Fig. 3c). Next, to analyze the extent to which global changes in H3K9me2 affect gene expression, we conducted RNA-seq analysis of RK-701−treated HUDEP-

2 cells (Fig. 3d), and identified 204 genes that were transcriptionally upregulated by RK-701 treatment (Supplementary Table 6). Selective induction of fetal and embryonic globins was observed by RNA-seq analysis (Fig. 3d and Supplementary Table 7). Significant upregulation of globin-related gene products (hemoglobin complex) was verified by gene ontology (GO) analysis (Supplementary Fig. 7a). However, among the previously identified globin switching regulators, including transcriptional factors, epigenetic factors, and non-coding RNAs[5], only BGLT3, a lncRNA encoded in the β-globin gene locus, was selectively upregulated by RK-701 (Fig. 3d and Supplementary Table 7). Likewise, RNA-seq analysis upon shRNA-directed G9a knockdown also showed the induction of fetal and embryonic globins, and BGLT3 in HUDEP-2 cells (Supplementary Fig. 7b and Supplementary Tables 7 and 8). BGLT3 lncRNA has been shown to function as a positive regulator of γ-globin gene expression[10]. Consistent with this, overexpression of BGLT3 promoted γ-globin expression in HUDEP-2 cells (Supplementary Fig. 8a and b). qPCR analysis confirmed that RK-701, but not RK-0133114, induced BGLT3 expression in both HUDEP-2 cells (Fig. 3e) and primary human CD34[+] cells (Fig. 3f) at concentrations similar to those that induced γ-globin and decreased H3K9me2 levels (Fig. 1f and Fig. 2a−e). G9a knockdown by shRNA phenocopied the RK-701-induced increase in both γ-globin and BGLT3 expression in HUDEP-2 cells (Fig. 3g−i). In addition, both UNC0638 (Supplementary Fig. 8c−e) and CM-272, a dual G9a/DNA methyltransferases (DNMTs) inhibitor (Supplementary Fig. 8f−h)[30], also induced the expression of both BGLT3 and γ-globin with a reduction of H3K9me2 levels in a concentration-dependent manner. These results suggest that G9a inhibition can selectively restore the BGLT3 expression that has been silenced in erythroid cells. Next, we used shRNA to evaluate the effect of BGLT3 knockdown on RK-701-induced γ-globin expression. Importantly, the induction of γ-globin by RK-701 was almost completely abolished in all three BGLT3-knockdown cell lines (Fig. 3j, k), suggesting that BGLT3 lncRNA is indispensable for γ-globin reactivation by G9a inhibition.

### Involvement of two HbF repressors in RK-701-induced γ-globin expression

To elucidate how G9a suppresses BGLT3 transcription, we examined the role of two HbF repressors BCL11A and ZBTB7A. BCL11A is known to be a central transcriptional factor that regulates globin switching[6,31], and has been shown to negatively regulate the transcription of BGLT3[32]. ZBTB7A (also known as LRF) has recently been identified as another potent repressor of γ-globin expression, acting independently of BCL11A[8,33]. Currently, however, it is unclear whether ZBTB7A is involved in the repression of BGLT3. We therefore examined whether the gene knockouts of these two HbF repressors affected RK-701−induced BGLT3 expression. As reported previously[32], BGLT3 expression was increased in BCL11A knockout HUDEP-2 cells (Fig. 4a). In the absence of BCL11A, RK-701 could no longer increase BGLT3 expression in HUDEP-2 cells (Fig. 4a). Similarly, BGLT3 expression was increased in the absence of ZBTB7A but ZBTB7A knockout prevented RK-701 from increasing BGLT3 expression in HUDEP-2 cells (Fig. 4a). We also tested the effects of BCL11A and ZBTB7A knockouts on RK-701-induced γ-globin expression in HUDEP-2 cells (Supplementary Fig. 9a). Consistent with previous observations[8], the expression of γ-globin, but

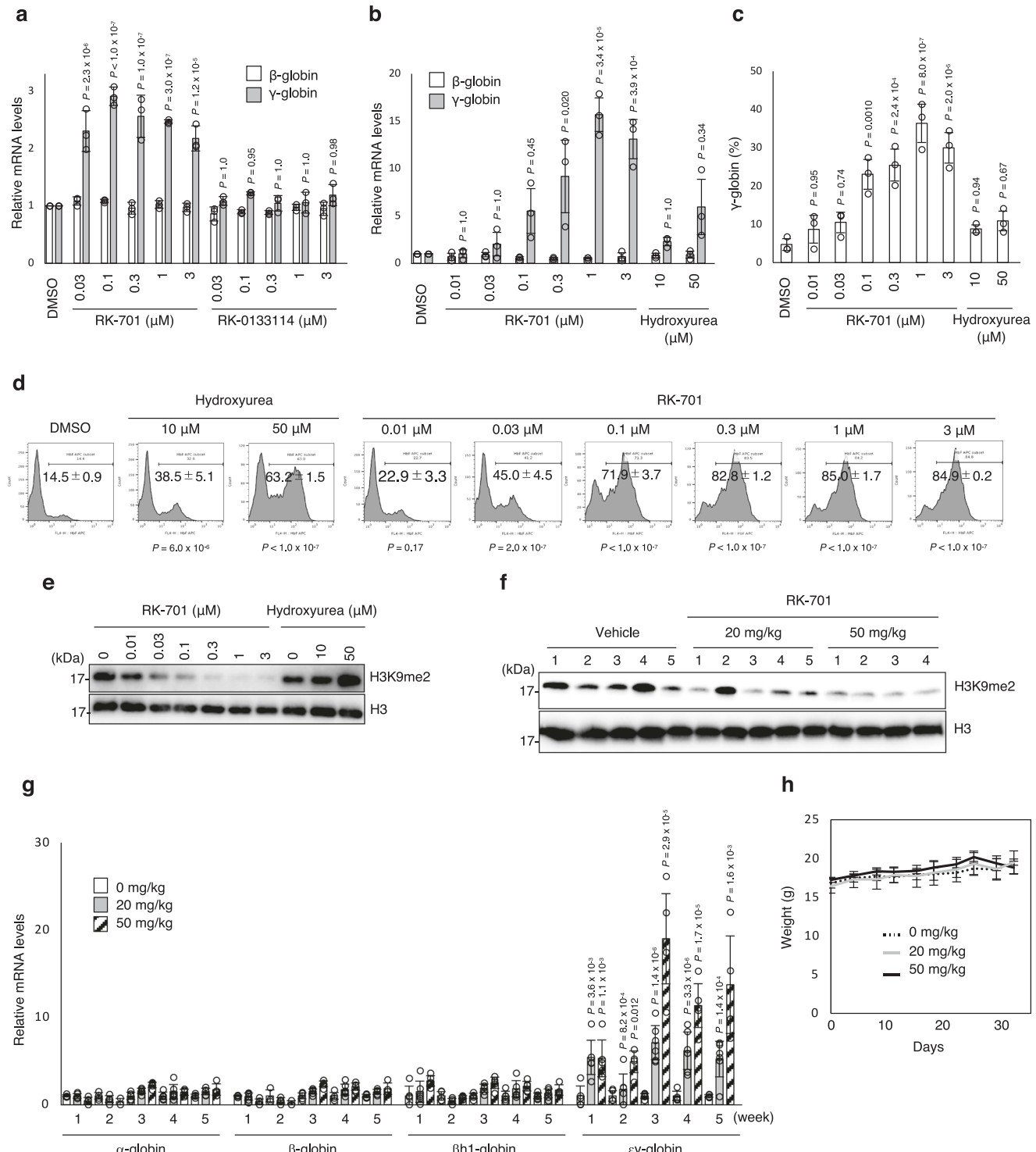

not β-globin, was dramatically increased in both BCL11A and ZBTB7A knockout cells. In the absence of BCL11A, γ-globin expression was slightly increased by treatment with RK-701, while ZBTB7A knockout cells did not respond to RK-701. These results suggest that BCL11A and ZBTB7A, two major transcriptional repressors that play pivotal roles in inhibiting the expression of both BGLT3 and γ-globin, are the major targets for reactivation of γ-globin gene expression by RK-701.

We next tested the possibility that BCL11A or ZBTB7A recruits G9a onto the BGLT3 gene region to enhance the H3K9me2 level. The ChIP-qPCR analysis demonstrated that the occupancy of H3K9me2 in the BGLT3 gene region was selectively and significantly reduced by

knocking out BCL11A or ZBTB7A genes (Fig. 4b). In addition, BCL11A and ZBTB7 physically interacted with G9a in cells (Fig. 4c, d). These observations suggest that BCL11A and ZBTB7 recruit G9a to the BGLT3 gene locus, where G9a-catalyzed H3K9me2 is facilitated.

Potential mechanisms of how RK-701 suppresses the function of BCL11A and ZBTB7 include inhibition of BCL11A and ZBTB7 gene expression and inhibition of their recruitment to the BGLT3 gene locus. RK-701 did not significantly affect the transcriptional levels of BCL11A or ZBTB7 (Fig. 3d, Supplementary Fig. 9b, and Supplementary Table 7). We also examined the effects of BCL11A and ZBTB7 on the H3K9me2 status in the gene loci. While ChIP-seq analysis showed that

**Fig. 2 | A specific G9a inhibitor RK-701 induces the expression of fetal globin.**
**a** Relative expression of β- and γ-globin genes in HUDEP-2 cells treated with various concentrations of RK-701 or RK-0133114 for 4 days. Amounts of β- and γ-globin RNAs were measured by RT-qPCR. Data are mean ± SD from three independent experiments. **b** Relative expression of β- and γ-globin mRNAs in CD34[+]-derived cells treated with RK-701 or hydroxyurea on day 10 of erythroid differentiation. Data are mean ± SD from three independent experiments. **c** The percentage of γ-globin proteins, relative to total β-like globin proteins (γ-globin + β-globin), in CD34[+]-derived cells treated with RK-701 or hydroxyurea on day 11 of erythroid differentiation. Data are mean ± SD from three independent experiments. **d** Flow cytometric analysis of γ-globin expression in CD34[+]-derived cells treated with RK-701 or hydroxyurea on day 11 of erythroid differentiation. Percentages of HbF-expressing cells were determined by gating HbF-positive cells. Data are mean ± SD from three independent experiments. A representative image of three different analyses is shown. **e** H3K9me2 levels in CD34[+] cells treated with RK-701 or hydroxyurea on day 10 of erythroid differentiation. A representative image of three different analyses is shown. Blots of H3 and H3K9me2 on different gels were derived from the same sample and processed in parallel. **f** H3K9me2 levels in peripheral blood of mice after treatment with RK-701 for 5 weeks ($n = 4$–5 mice per group). Blots of H3 and H3K9me2 on different gels were derived from the same sample and processed in parallel. **g** Fold changes in α-, β-, εγ-, and βh1-globin mRNA levels relative to *Gapdh* in peripheral blood of mice after treatment with RK-701 for 1, 2, 3, 4, and 5 weeks ($n = 3$–6 mice per group). **h** Mice were intraperitoneally administered RK-701 over 5 weeks, within each week for 5 consecutive days followed by 2 days off. The body weight of individual mice was measured every 3 or 4 days ($n = 4$–6 mice per group). *P*-value was calculated by the one-way ANOVA with Tukey's post hoc test except for (**g**) calculated by the one-sided Williams test.

H3K9me2 levels at the BCL11A gene locus were slightly decreased in RK-701–treated cells (Supplementary Fig. 9c), the H3K9me2 occupancy at the ZBTB7A gene locus was unchanged (Supplementary Fig. 9d), indicating that the expression of BCL11A and ZBTB7 is essentially unaffected by RK-701. Consistent with the previous observation that BCL11A binds to the intergenic region between *HBD* and *HBG1* in the β-globin gene locus where BGLT3 gene is located (Fig. 3b)[7], ChIP-qPCR revealed that BCL11A was significantly associated with the BGLT3 gene region (Fig. 5a). It is particularly worth noting that occupancies of BCL11A in the BGLT3 gene region were significantly suppressed by RK-701 (Fig. 5a), suggesting the involvement of histone H3K9me2 in BCL11A binding to the region. Similarly, ZBTB7A also bound to the BGLT3 gene region, and its occupancy in this region was significantly reduced by RK-701 (Fig. 5b). The peptide pull-down assay showed that the binding of BCL11A and ZBTB7A to histone H3 peptides was increased by H3K9 methylation (Fig. 5c), suggesting that G9a enhances the recruitment of the complex containing BCL11A or ZBTB7A.

Histone H3K9 methylation is generally recognized by reader proteins containing conserved protein domains such as the chromodomain and the PHD domain. Therefore, it seems likely that the recruitment of these two repressors is mediated by such reader proteins for H3K9me2. Both BCL11A and ZBTB7A are known to interact with the NuRD complex[8,13,14], in which CHD3 and CHD4 are components that can bind to histone H3 methylated at K9 through their PHD domain[34,35]. To test whether CHD3 or CHD4 is involved in the repressor recruitment, we firstly examined the effect of RK-701 on the occupancy of CHD3 and CHD4 on the BGLT3 gene locus (Fig. 5d). ChIP-qPCR analyses revealed that CHD4 was highly associated with the BGLT3 gene region and that the occupancy of CHD4, but not CHD3, on the BGLT3 gene locus was significantly decreased in RK-701–treated cells. Consistent with previous reports[36,37], we confirmed that CHD4 interacted with BCL11A and ZBTB7A (Fig. 5e, f) and bound to histone H3 peptides methylated at K9 (Fig. 5c). Thus, two repressors may congregate at the BGLT3 gene locus by interacting with the methylation reader protein CHD4, but the G9a/GLP inhibition by RK-701 suppresses the CHD4-mediated recruitment of BCL11A and ZBTB7A to the BGLT3 gene locus.

To investigate whether BGLT3 regulates γ-globin expression downstream of BCL11A and ZBTB7A, we suppressed the expression of BGLT3 in BCL11A and ZBTB7A knockout HUDEP-2 cells using shRNA-based knockdown. BGLT3 knockdown attenuated the increase in γ-globin expression in both BCL11A and ZBTB7A knockout cells (Fig. 6a, b). On the other hand, overexpression of BGLT3 did not impair BCL11A and ZBTB7A expression (Fig. 6c) despite increased γ-globin expression (Supplementary Fig. 8b). Thus, BGLT3 appears to act downstream of both BCL11A and ZBTB7A to induce HbF. Together, our findings suggest that two HbF repressors in complex with G9a are recruited to the BGLT3 gene locus depending on histone H3K9me2, which further propagates H3K9me2, resulting in the congregation of repressor complexes (see Discussion). On the other hand, inhibition of G9a/GLP by RK-701 dissociates the repressive complex and increases BGLT3 expression, which in turn leads to the upregulation of γ-globin expression. Given that the levels of γ-globin induction achieved by BCL11A or ZBTB7A knockout are higher than those caused by RK-701 (Supplementary Fig. 9a), it seems likely that BCL11A and ZBTB7A suppress γ-globin expression through both G9a/GLP-dependent and -independent gene silencing mechanisms. In support of this hypothesis, only a modest overlap in gene expression changes was observed between RK-701-treated and BCL11A or ZBTB7A knockout cells, although both γ-globin and BGLT3 genes are commonly upregulated (Supplementary Fig. 9e and Supplementary Table 8).

## The pivotal role of BGLT3 in pharmaceutical HbF induction
To investigate whether BGLT3 plays a universal role in the control of γ-globin expression beyond G9a/GLP-mediated regulatory pathways, we next determined the expression of BGLT3 following hydroxyurea treatment. Although hydroxyurea has been used for the clinical treatment of SCD for more than 20 years, how it induces HbF is still unknown. Like RK-701, hydroxyurea promoted BGLT3 expression in both HUDEP-2 (Supplementary Fig. 10a) and human CD34[+] hematopoietic cells (Supplementary Fig. 10b). Importantly, hydroxyurea–induced upregulation of γ-globin was significantly suppressed by BGLT3 knockdown (Fig. 7a, b), suggesting that reactivation of γ-globin expression by hydroxyurea is also mediated by BGTL3. Although we showed that BGLT3 expression was suppressed by BCL11A and ZBTB7A (Fig. 4a), hydroxyurea did not significantly affect the expression of either BCL11A or ZBTB7A (Supplementary Fig. 10c). In addition, hydroxyurea failed to further enhance the expression of BGTL3 and γ-globin, whose levels were already high in BCL11A and ZBTB7A knockout HUDEP-2 cells (Fig. 7c and Supplementary Fig. 10d). In contrast to RK-701, however, hydroxyurea did not alter H3K9me2 levels at the β-globin gene locus (Supplementary Fig. 10e). Nevertheless, ChIP analyses revealed that occupancies of BCL11A at the β-globin gene locus, including the HS2 and BGLT3 gene regions, were decreased by hydroxyurea, while those of ZBTB7A were unchanged (Fig. 7d, e). These observations suggest that BGLT3 lncRNA is also critical for hydroxyurea-induced γ-globin expression, although its mechanism is independent of H3K9me2 levels at the β-globin gene locus. This observation prompted us to further investigate the involvement of BGLT3 in γ-globin induction by other epigenetic modulators such as decitabine, a DNMT inhibitor, and MS-275 and SAHA, both of which are histone deacetylase (HDAC) inhibitors[14,18]. BGLT3 expression was enhanced by all these HbF inducers (Supplementary Fig. 11a), and BGLT3 knockdown suppressed the induction of γ-globin expression (Supplementary Fig. 11b–d). The ChIP-qPCR analysis confirmed that histone H3 acetylation at the β-globin gene locus including the BGLT3 gene region was increased by SAHA in HUDEP-2 cells (Supplementary Fig. 11e). Treatment with both RK-701 and decitabine significantly increased the expression of BGLT3 and γ-globin compared with each

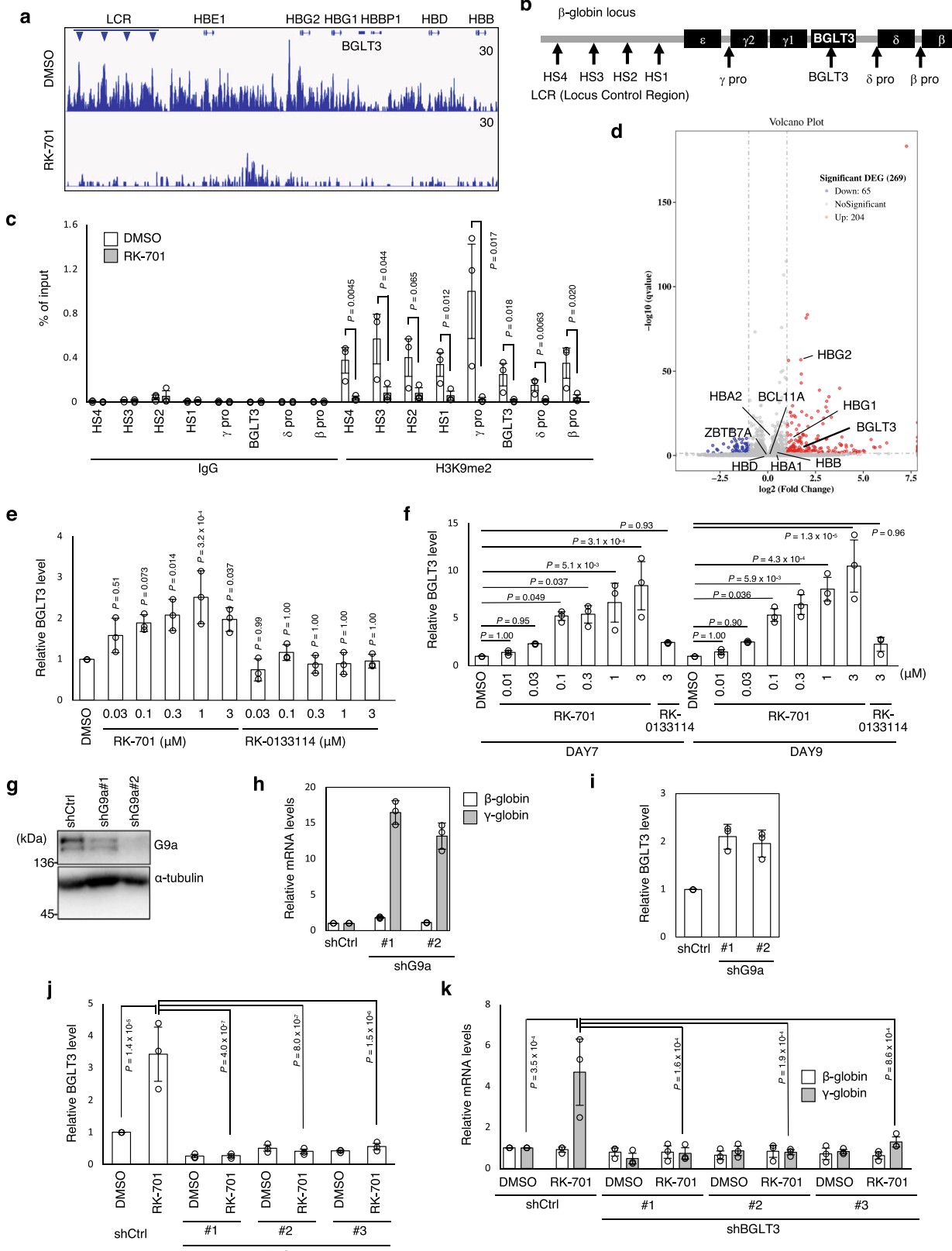

compound alone (Supplementary Fig. 11f and g). Notably, the expression levels of γ-globin are well correlated with those of BGLT3. These results suggest that BGTL3 expression is required for the induction of γ-globin gene expression by all these epigenetic regulators. Finally, we investigated the general relevance of BGLT3 to γ-globin expression using HiDEP-1 cells, a human erythroid progenitor cell line derived

from iPS cells, in which γ-globin is constantly expressed at a much higher steady-state level than in HUDEP-2 cells (Fig. 7f)[38]. Consistent with the high-level expression of γ-globin, the level of BGLT3 expression was much higher in HiDEP-1 cells than in HUDEP-2 cells (Fig. 7f), and BGLT3 knockdown abolished the expression of γ-globin in HiDEP-1 cells (Fig. 7g). All of these results suggest that BGLT3 lncRNA is the

**Fig. 3 | The essential role of BGLT3 in induction of γ-globin by RK-701. a** ChIP-seq traces for H3K9me2 on the β-globin locus in HUDEP-2 cells after 4-day treatment with 1 μM RK-701 or DMSO. A representative image of two independent experiments is shown. **b** Diagram of the human β-globin locus. Positions of amplicons used for ChIP-qPCR are indicated. **c** ChIP-qPCR analyses using antibodies against IgG or H3K9me2 in HUDEP-2 cells treated with 1 μM RK-701 or DMSO for 4 days. Data are mean ± SD from three independent experiments. **d** A volcano plot illustrating changes in gene expression induced by RK-701. RNA-seq analysis was performed in HUDEP-2 4 days after treatment with 1 μM RK-701 or DMSO. The plot represents statistical significance versus fold change in gene expression between the two conditions. Results from three biological replicates are shown. Red, gray, or blue dots indicate differentially expressed genes (DEG) for upregulated, not significant or downregulated. Some of the data points that are out of scale indicate the value of inf (infinite) in Supplementary Table 6. **e** Relative expression of BGLT3 RNA in HUDEP-2 cells treated with RK-701 or RK-0133114 for 4 days. Data are mean ± SD from three independent experiments. **f** Relative expression of BGLT3 RNA in CD34[+] cells treated with RK-701 or hydroxyurea on day 7 or day 9 of erythroid differentiation. Data are mean ± SD from three independent experiments. **g–i** Effects of G9a knockdown on γ-globin and BGLT3 gene expression in HUDEP-2 cells. G9a knockdown efficiency was evaluated by western blotting (**g**). A representative image of two independent experiments is shown. Relative expression of β- and γ-globin (**h**) and BGTL3 (**i**) in HUDEP-2 cells was analyzed by RT-qPCR. Data are mean ± SD from three independent experiments. **j, k** Effects of BGLT3 knockdown on the RK-701–induced expression of BGLT3 or γ-globin. Relative expression of BGLT3 (**j**) or γ-globin (**k**) in BGLT3-knockdown HUDEP-2 cells treated with 1 μM RK-701 for 4 days was analyzed by RT-qPCR. Data are mean ± SD from three independent experiments. *P*-value was calculated by the one-way ANOVA with Tukey's post hoc test.

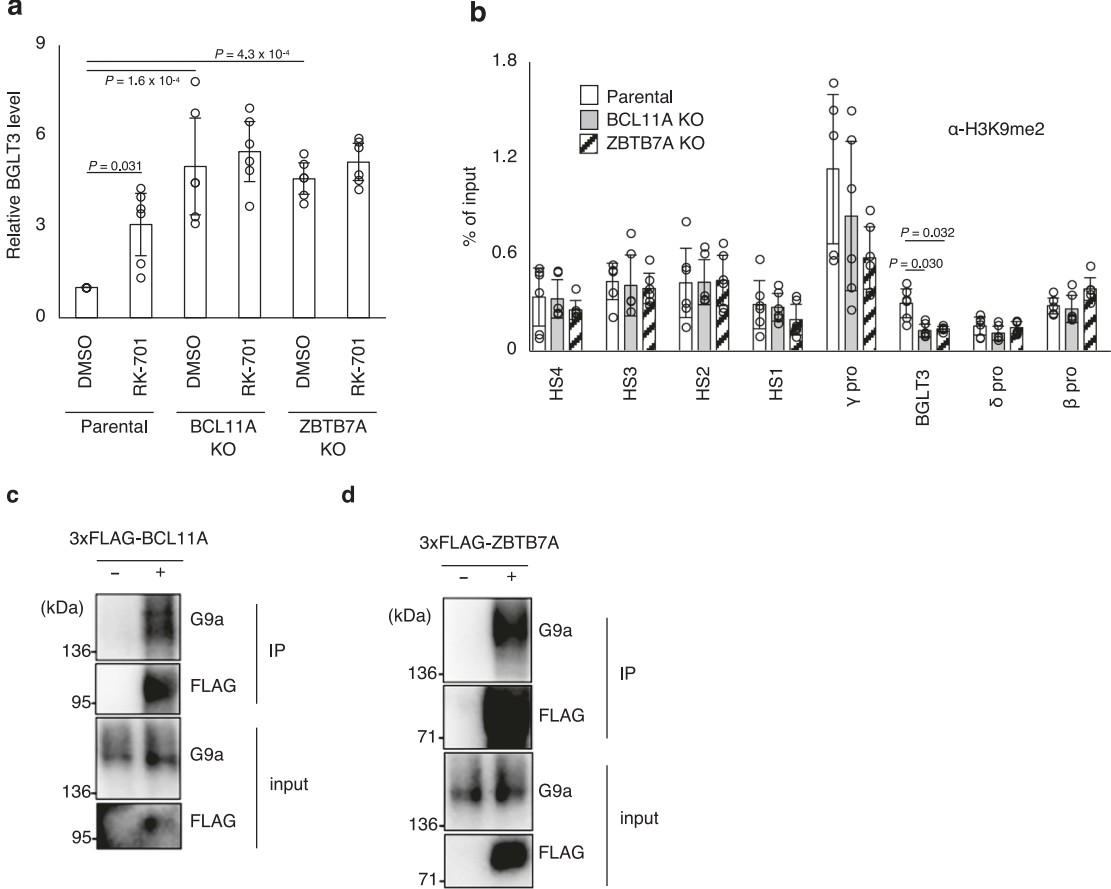

**Fig. 4 | Regulation of G9a-induced H3K9me2 at the BGLT3 gene locus by BCL11A and ZBTB7A. a** Effect of BCL11A or ZBTB7A knockout on RK-701–induced expression of BGLT3. Relative expression of BGLT3 in parental, BCL11A-, or ZBTB7A-knockout HUDEP-2 cells treated with 1 μM RK-701 for 4 days was analyzed by RT-qPCR. Data are mean ± SD from three independent experiments. **b** ChIP-qPCR analyses using an antibody against H3K9me2 in parental, BCL11A-, or ZBTB7A-knockout HUDEP-2 cells. Data are mean ± SD from three independent experiments. **c, d** Interaction between G9a and BCL11A (**c**) or ZBTB7A (**d**) in cells. HEK293T cells were transiently transfected with plasmids for expression of FLAG-tagged BCL11A or ZBTB7A. Proteins immunoprecipitated with the anti-FLAG antibody were analyzed by immunoblotting using the indicated antibodies. A representative image of three independent experiments is shown. *P*-value was calculated by the one-way ANOVA with Tukey's post hoc test (**a**) or the two-tailed *t*-test (**b**).

general activator essential for the pharmacological induction of γ-globin gene transcription.

## Discussion

Small-molecule compounds that can induce HbF are expected to be therapeutic agents for SCD. Small-molecule inhibitors that target epigenetic enzymes, such as G9a and GLP, are candidate therapeutic HbF inducers. Indeed, studies aimed at developing epigenetic modulators for SCD treatment have been highly successful[39–41]. In this study, we developed RK-701, a specific and essentially non-toxic G9a/GLP

inhibitor with a chemical scaffold that differs from that of UNC0638. We showed that RK-701 induced fetal globin gene expression in both human erythroid cells and mice with better efficacy than hydroxyurea, an existing SCD therapeutic agent (Fig. 2).

Hydroxyurea has long been used to treat SCD, but its toxicity is one of its major clinical drawbacks[2]. In the present study, hydroxyurea-induced HbF expression but also caused severe cytotoxicity in primary human CD34[+] hematopoietic cells (Supplementary Fig. 5g). In addition, hydroxyurea has been reported to be mutagenic and genotoxic, probably due to its inhibition of ribonucleotide reductase, which is

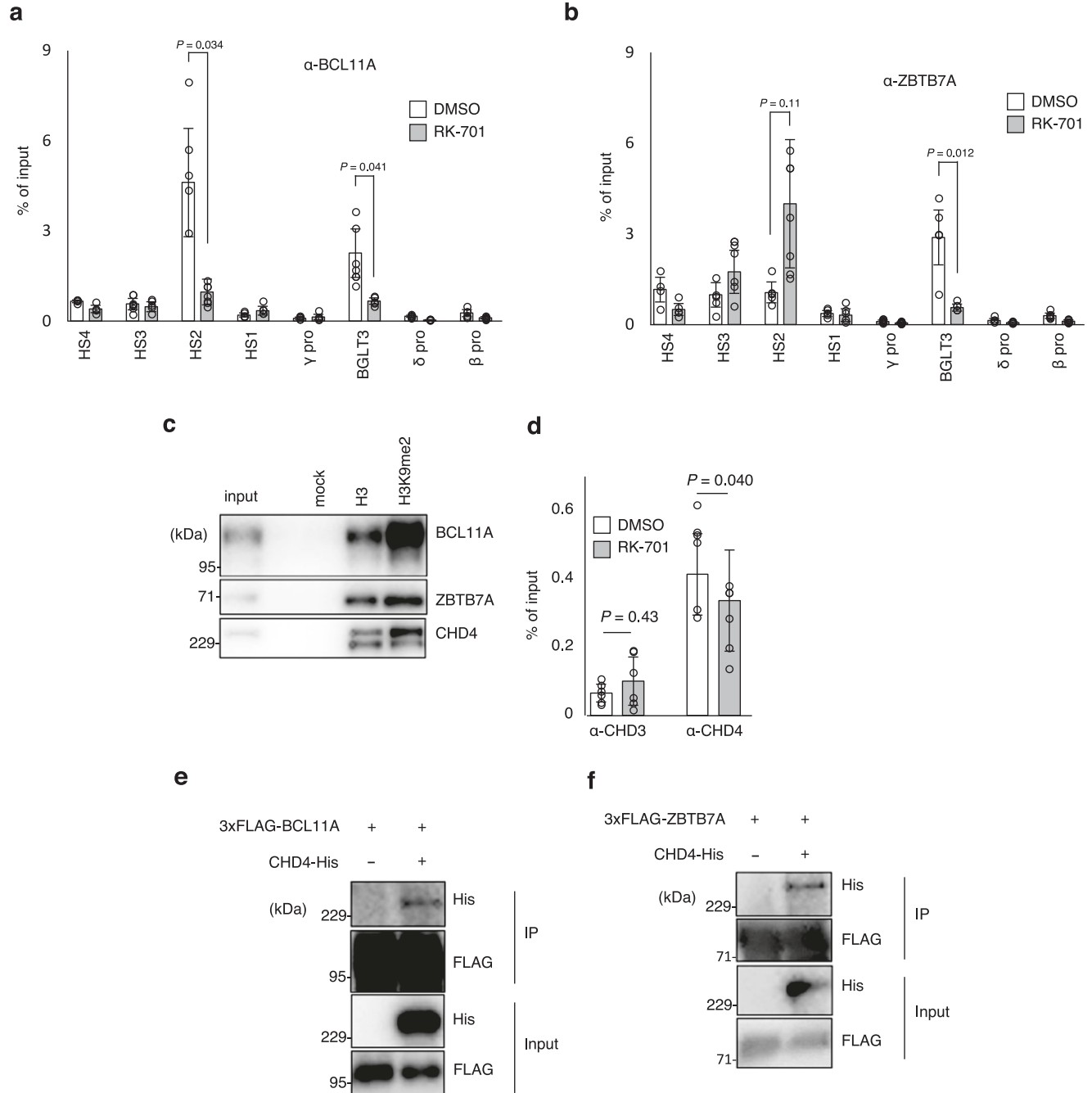

**Fig. 5 | Regulation of occupancies of BCL11A and ZBTB7A at the BGLT3 gene locus by G9a. a**, **b** ChIP-qPCR analyses using antibodies against BCL11A (**a**) and ZBTB7A (**b**) in HUDEP-2 cells treated with 1 μM RK-701 or DMSO for 4 days. Data are mean ± SD from three independent experiments. **c** Peptide pull-down assay with biotinylated histone H3 peptides from HUDEP-2 cell lysates. A Representative image of two independent experiments is shown. **d** ChIP-qPCR analyses using antibodies against CHD3 or CHD4 in HUDEP-2 cells treated with 1 μM RK-701 or DMSO for 4 days. Data are mean ± SD from three independent experiments. **e**, **f** Interaction between CHD4 and BCL11A (**e**) or ZBTB7A (**f**) in cells. HEK293T cells were transiently transfected with plasmids for expression of His-tagged CHD4 and FLAG-tagged BCL11A or ZBTB7A. Proteins immunoprecipitated with the anti-FLAG antibody were analyzed by immunoblotting using the indicated antibodies. A representative image of two independent experiments is shown. *P*-value was calculated by two-tailed *t*-test.

required for DNA synthesis[42]. In contrast, RK-701 exhibited low cytotoxicity and no mutagenicity or genotoxicity (Supplementary Fig. 4a, b, e, and f, and Supplementary Fig. 5g). Thus, RK-701 seems to have better efficacy and safety than hydroxyurea. Since it also has excellent specificity and pharmacokinetics, RK-701 is a promising lead compound for the development of therapeutic agents for SCD. In addition, its remarkably lower toxicity than most commonly used G9a inhibitors, such as UNC0638, and the fact that the inactive *R*-enantiomer RK-

0133114 is an ideal control, indicate that RK-701 is the best chemical tool thus far for investigating the physiological roles of G9a both in vitro and in vivo.

A series of experiments with RK-701 uncovered the molecular mechanism by which G9a/GLP regulates γ-globin gene expression in erythroid cells. Our data suggest that BCL11A and ZBTB7A promote H3K9me2 by recruiting G9a to the BGLT3 gene locus, resulting in suppression of BGLT3 lncRNA transcription and subsequent γ-globin

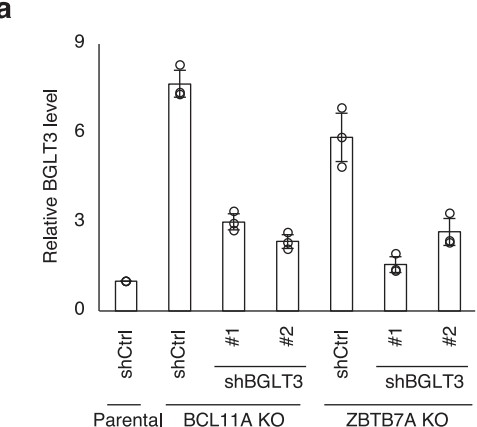

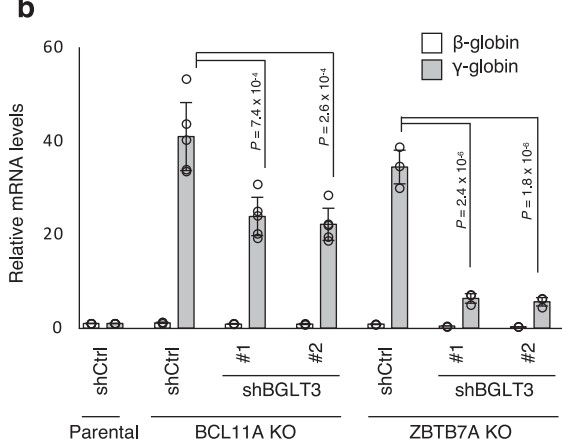

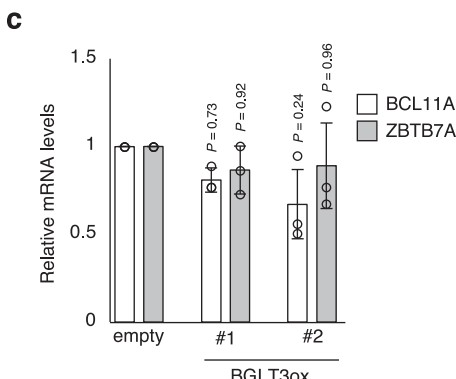

**Fig. 6 | The role of BGLT3 in γ-globin induction by BCL11A and ZBTB7A.**
**a** Validation of shRNA-mediated knockdown efficiency of BGLT3 in BCL11A- or ZBTB7A-knockout HUDEP-2 cells. Data are mean ± SD from three independent experiments. **b** Relative expression of γ-globin in BCL11A- or ZBTB7A-knockout HUDEP-2 cells with or without BGLT3 knockdown. Data are mean ± SD from three

independent experiments. **c** Relative expression of BCL11A and ZBTB7A in BGLT3-overexpressing HUDEP-2 cells. Data are mean ± SD from three independent experiments. P-value was calculated by the one-way ANOVA with Tukey's post hoc test.

expression. Because BCL11A and ZBTB7A binding motifs were present in the BGLT3 gene region (Supplementary Fig. 12) and both repressors bound to the BGLT3 gene region (Fig. 5a, b), these repressors may repress the transcription of the BGLT3 gene by facilitating locus-specific H3K9me2 by G9a. In addition, we showed that G9a-induced H3K9me2 promotes the recruitment of BCL11A and ZBTB7A to the BGLT3 gene locus at least in part via interacting with CHD4. This locus-specific recruitment of these repressors may propagate the H3K9me2 state of chromatin by a positive feedback loop mechanism to further recruit G9a with BCL11A/ZBTB7A to the BGLT3 gene locus, which ensures the repression state of BGLT3 and the subsequent silencing of γ-globin (Fig. 8). Alternatively, these repressors may be recruited to the BGLT3 gene locus by altering chromatin structures because G9a modulates chromatin accessibility[43]. In support of this hypothesis, it was reported that G9a inhibition alters chromosomal loop formation in the β-globin gene locus[21]. It also seems possible that BCL11A, but not ZBTB7A, indirectly regulates *BGLT3* gene expression through dynamic chromatin rearrangement, because the occupancy of BCL11A on the HS2 region in the LCR was decreased by RK-701 (Fig. 5a). Previous reports indicated that BCL11A binds to the HS2 region in the LCR and suppresses γ-globin transcription by reconstituting the β-globin cluster through regulation of chromosomal loop formation[7,32]. By drastically decreasing the occupancy of BCL11A on HS2, RK-701 might indirectly contribute to enhancing the expression of BGLT3 through a specific chromatin architecture.

DNA methylation may also be involved in RK-701-induced HbF expression because a recent report suggests that G9a plays a role in

maintaining DNA methylation[44]. Indeed, decitabine, a DNMT inhibitor, induced γ-globin expression (Supplementary Fig. 11a and f)[22]. Interestingly, the combination of RK-701 with decitabine synergistically increased both BGLT3 and γ-globin gene expression (Supplementary Fig. 11f and g). These results suggest that combinations of different epigenetic inhibitors represent a useful strategy for the treatment of SCD.

Investigation of the mode of action of RK-701 revealed that BGLT3 lncRNA, acting downstream of BCL11A and ZBTB7A, played a critical role in reactivating HbF in erythroid cells. Although several mechanisms by which hydroxyurea induces HbF have been proposed[45], this issue remains a long-standing mystery. In this study, we found that hydroxyurea-induced γ-globin using machinery very similar to that of RK-701, i.e., hydroxyurea produced γ-globin by promoting BGLT3 expression. In contrast to RK-701, however, hydroxyurea did not affect the H3K9me2 level (Supplementary Fig. 10e). By decreasing the occupancy of BCL11A at the HS2 and BGLT3 gene regions (Fig. 7d), hydroxyurea may promote BGLT3 transcription in a H3K9me2-independent manner. It is assumed that hydroxyurea induces HbF activity by remodeling the chromatin structure via DNA replication stress caused by ribonucleotide reductase inhibition[46]. Therefore, this chromatin structural change may cause BGLT3 expression by preventing BCL11A recruitment without affecting H3K9me2 levels. Interestingly, other epigenetic modulators such as a DNMT inhibitor and HDAC inhibitors also promoted BGLT3 expression, and their HbF induction depended at least in part on BGLT3 lncRNA (Supplementary Fig. 11b–d). Thus,

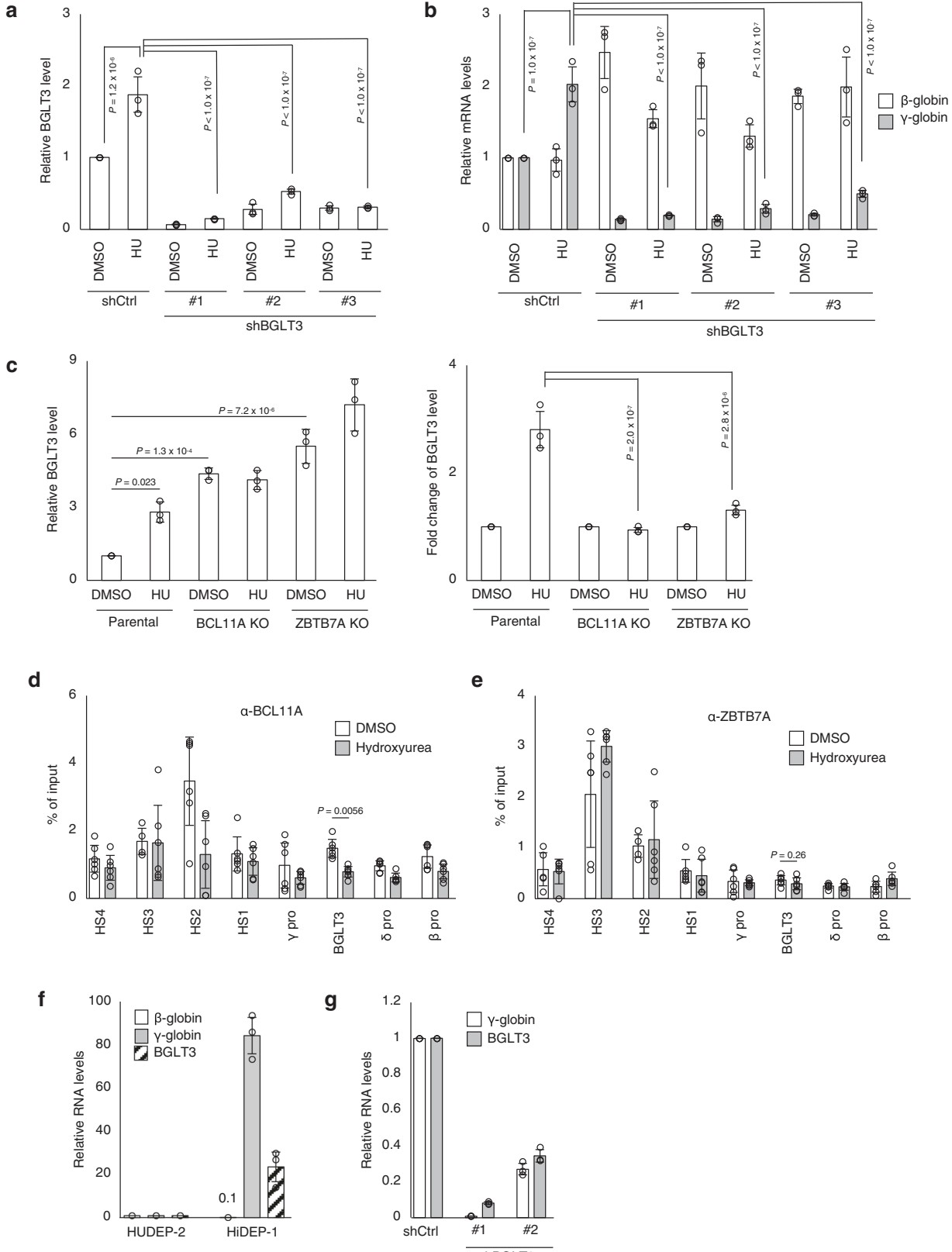

BGLT3 lncRNA appears to play a key role in HbF induction by various small molecules.

Even in HiDEP-1 cells, which maintained high γ-globin expression (Fig. 7f), this expression was completely dependent on BGLT3 (Fig. 7g), suggesting that the importance of BGLT3 is not limited to pharmacological γ-globin induction. Gene expression profiling during

development has shown that both BGLT3 and γ-globin are co-expressed during the fetal stage[10]. In primates, simians have highly conserved homologs with sequence homology to the human BGLT3 gene, but this is not the case for other mammals such as prosimians (Supplementary Fig. 13). Simians have γ-globins that are expressed during the fetal stage, whereas prosimians and other species do not

**Fig. 7 | The role of BGLT3 as the universal regulator in γ-globin induction.**
**a, b** Effects of BGLT3 knockdown on the hydroxyurea–induced expression of
BGLT3 or γ-globin. Relative expression of BGLT3 (**a**) or γ-globin (**b**) in BGLT3-
knockdown HUDEP-2 cells treated with 200 μM hydroxyurea for 4 days was ana-
lyzed by RT-qPCR. Data are mean ± SD from three independent experiments.
**c** Relative expression of BGLT3 in parental, BCL11A-, or ZBTB7A-knockout HUDEP-2
cells after 4-day treatment with 200 μM hydroxyurea. Right panel in (**c**) shows
BGLT3 gene expression relative to DMSO control in each cell line. Data are

mean ± SD from three independent experiments. **d, e** ChIP-qPCR analyses using
antibodies against BCL11A (**d**), or ZBTB7A (**e**) in HUDEP-2 cells treated with 200 μM
hydroxyurea or DMSO for 4 days. Data are mean ± SD from three independent
experiments. **f** Relative expression of β- and γ-globin and BGLT3 in HUDEP-2 and
HiDEP-1 cells. Data are mean ± SD from three independent experiments. **g** Relative
expression of γ-globin in BGLT3-knockdown HiDEP-1 cells. Data are mean ± SD from
three independent experiments. *P*-value was calculated by the one-way ANOVA
with Tukey's post hoc test except for (**d**) and (**e**), calculated by two-tailed *t*-test.

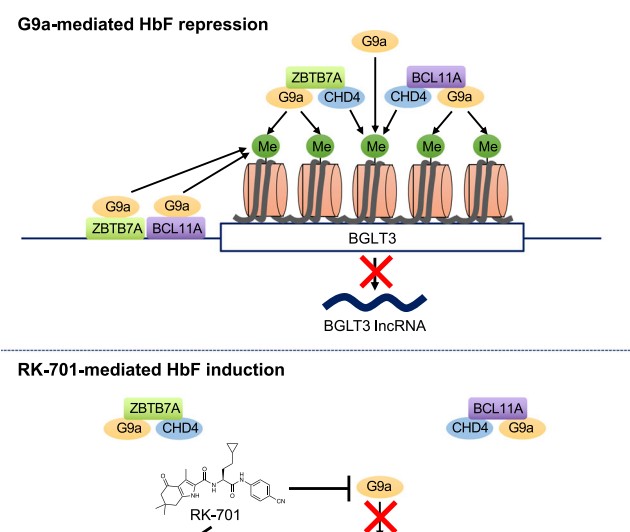

**Fig. 8 | A schematic model of the molecular mechanism by which G9a represses
fetal globin gene expression.** Two major HbF repressors, BCL11A and ZBTB7A,
recruit G9a onto the BGLT3 gene locus, where G9a facilities H3K9me2. G9a-
mediated H3K9me2 recruits BCL11A and ZBTB7A onto the BGLT3 gene locus via
CHD4, which may further enhance locus-specific H3K9me2. This positive feedback
loop of enhancement of H3K9me2 at the BGLT3 gene locus ensures to repress
BGLT3 lncRNA and subsequent fetal globin expression. RK-701 reactivates fetal
globin by disrupting this G9a–BCL11A/ZBTB7A system by inhibiting the enzymatic
activity of G9a.

(Supplementary Fig. 14). It appears that only species harboring BGLT3
homologs possess fetal globins, and that these proteins have coe-
volved in primates. All of these observations clearly indicate that
BGLT3 is the universal activator for fetal globin gene expression. The
essential role of BGLT3 lncRNA in the induction of HbF highlights its
promise as a therapeutic target for SCD.

## Methods
### Cell culture
HUDEP-2 cells and HiDEP-1 cells were grown in Stemline II Hemato-
poietic Stem Cell Expansion Medium (Sigma) supplemented with
50 ng/mL recombinant stem cell factor (SCF; R&D systems), 3 IU/mL
recombinant erythropoietin (EPO; Kyowa Kirin), 1 μM dexamethasone
(Dex; Sigma), and 1 μg/mL doxycycline (Sigma). A total of $1 \times 10^5$
human CD34$^+$ progenitor cells derived from bone marrow or periph-
eral blood (Lonza) were cultured on 35-mm dishes (BD Falcon) in
StemSpan SFEMII medium with 50 ng/mL SCF, 5 IU/mL EPO, and 10 ng
/mL interleukin 3 (IL-3; R&D systems). The medium was changed on
day 4 to SCF, EPO, and 1 μM Dex, and the cells were transferred to 60-

mm dishes (BD Falcon). Seven days after the onset of differentiation,
cells were counted and $1 \times 10^6$ cells were re-plated on 100-mm dishes
(BD Falcon) with SCF, EPO, and Dex. The rat myoblast cell line H9c2,
HEK293T, and PLAT-E cells were maintained in Dulbecco's modified
Eagle's medium (Thermo Fisher Scientific) supplemented with 10%
heat-inactivated fetal bovine serum (FBS) and penicillin/streptomycin.
All cells were cultured at 37 °C under 5% CO$_2$.

### High-throughput screening (HTS)
HTS was conducted in the chemical library at the Drug Discovery
Initiative (DDI) at The University of Tokyo by using a peptidyl-7-
amino-4-methylcoumarin amide (MCA) with a sequence derived
from histone H3 as a fluorogenic substrate (Supplementary
Table 9)[24]. Briefly, a recombinant mG9a protein (50 μg/mL) was pre-
incubated with each tested compound (4 or 20 μM) in HMT buffer
(50 mM Tris-HCl [pH 8.5,] 10 mM MgCl$_2$, 20 mM KCl, 250 mM
sucrose, 3 μg/mL BSA, and 10 mM 2-mercaptoethanol) for 1 h at room
temperature. Then, peptidyl-MCA substrate (60 μM Ac-ARK-MCA)
and SAM (1 mM) were added to the reaction mixture, and the mixture
was further incubated for 1 h at 37 °C. After incubation with trypsin
(10 mg/mL) at 37 °C for 15 min, the fluorescence intensity was mea-
sured at λex 330 nm/λem 380 nm.

### In vitro G9a methyltransferase assay
An amplified luminescence proximity homogeneous assay (ALPHA)
was performed with a modified version of a previously described
method[24]. Recombinant G9a proteins (BPS Bioscience) were incubated
with 15 μM *S*-adenosyl-methionine (Sigma) and 50 nM biotinylated
histone H3 (1-21; AnaSpec) in the assay buffer (50 mM Tris-HCl (pH
9.0), 50 mM NaCl, 0.01% Tween-20, and 1 mM DTT). After incubation at
room temperature for 1 h, AlphaLISA anti-H3K9me2 acceptor beads
(final concentration 10 μg/mL, PerkinElmer, AL117C) and AlphaScreen
streptavidin donor beads (final concentration 10 μg/mL, PerkinElmer,
6760002B) were added and incubated for an additional 1 h prior to
detection of the alpha signal with an EnSpire Alpha plate reader (Per-
kinElmer). ALPHA profile was analyzed using Origin (OriginLab).

### Kinetic analysis
Kinetic analysis was measured by ALPHA. Recombinant G9a proteins
(final concentration 0.0125 nM) were incubated with RK-701, a bioti-
nylated histone H3-derived peptide (final concentration: 6.25, 12.5, 25,
50, 100 nM) and SAM (final concentration 15 μM) in 10 μL of assay
buffer. In case of SAM plot, recombinant G9a proteins (final con-
centration 0.0125 nM) were incubated with RK-701, a biotinylated
histone H3-derived peptide (final concentration: 50 nM) and SAM (final
concentration 0.47, 0.94, 1.88, 3.75, 7.5, 15 μM) in 10 μL of assay buffer.
RK-701 was tested in 8-dose IC50 mode with 4-fold serial dilutions.
Data were analyzed using Excel and GraphPad Prism software (version
5.04) for IC50 curve fits using log (RK-701) versus response-variable
slope (four parameters) equation. ALPHA profile was analyzed using
Origin (OriginLab).

### Selectivity of RK-701
Selectivity of RK-701 against related methyltransferases, including
lysine methyltransferases, arginine methyltransferases, and DNA

methyltransferases, was evaluated by radioisotope-based methyltransferase assays (Reaction Biology). SAH or chaetocin was used as positive controls. RK-701 was tested in duplicate in a five-dose IC50 mode with 10-fold serial dilution, starting at 10 or 100 μM. SAH (S-(5′adenosyl-L-homocysteine) and chaetocin were tested as control compounds in 10-dose IC50 mode with three-fold serial dilution starting at 100 or 200 μM. Reactions were carried out at 1 μM SAM.

## Western blotting

Preparation of cell lysates and western blotting were performed as described previously[47]. Protein from cells lysates in lysis buffer (50 mM Tris-HCl (pH 7.5), 150 mM NaCl, 5 mM EDTA, 0.1% NP-40, 1 mM dithiothreitol) were separated by sodium dodecyl sulfate–polyacrylamide gel electrophoresis (SDS-PAGE) and transferred to a polyvinylidene difluoride (PVDF) membrane (Millipore) by electroblotting. The membranes incubated with primary antibodies overnight at 4 °C and secondary antibodies for 1 h at room temperature. The immune complexes were detected with the Immobilon Western Chemiluminescent HRP Substrate (Millipore). Luminescence was analyzed on a FUSION SOLO S (Vilber). Western blot bands were visualized using Evolution Capt software (M&S Instruments Inc.). Primary antibodies used were anti-G9a (1:1000, CST, #3306), anti-α-tubulin (1:5000, Abcam, ab18251), anti-H3K9me2 (1:3000, Abcam, ab1220), anti-H3 (1:5000, Abcam, ab1791), anti-H3K27me3 (1:3000, Sigma, 07-449), anti-GLP (1:1000, CST, #35005), anti-FLAG (100 ng/ml, Fujifilm, 01422383), anti-His (1:5000, MBL, PM032), anti-BCL11A (1:1000, abcam, ab19487), anti-ZBTB7A (1:1000, CST, #50565), anti-CHD3 (1:1000, Novus Biologicals, NB100-60412), and anti-CHD4 (1:1000, abcam, ab240640). Uncropped scans of all of the blots are available in Source Data file.

## Quantitative PCR (qPCR)

Total cellular RNA was prepared using the Tissue Total RNA Mini Kit (FAVORGEN), and cDNA was synthesized using ReverTra Ace Master Mix (TOYOBO). Expression of target genes was analyzed using THUNDERBIRD SYBR qPCR Mix (TOYOBO) on a CFX Connect Real Time System (Bio-Rad). The qPCR data were acquired on CFX Maestro (Bio-Rad). Primers are listed in Supplementary Table 10. *GAPDH* was used as a reference. Primers were purchased from INTEGRATED DNA TECHNOLOGIES or FASMAC.

## Aqueous solubility ([pH 7.4] and [pH 6.8])

The direct UV kinetic solution method was performed in duplicate. A 10 mM DMSO solution of compound was added to a microtube containing aqueous pH 7.4 or pH 6.8 phosphate buffer solution (maximum concentration: 200 μM). The solution mixture was vigorously shaken at ambient temperature for 1 h. The precipitate was removed by centrifugal separation, and the UV absorbance of the supernatant and the standard solution prepared by the test compound for calibration were measured in duplicate using a 96-well UV plate reader, SPECTRA MAX190 (Molecular Devices).

## Parallel artificial membrane permeability assay (PAMPA)

Parallel artificial membrane permeability measurements were performed in duplicate with a Corning Gentest Pre-coated PAMPA Plate System according to the standard protocol of Corning Incorporated Life Sciences (Tewksbury).

## Liver microsomal stability assay

[Human/Rat] Liver microsome analyses (human, rat) were performed in triplicate by the Sumika Chemical Analysis Service (SCAS) according to the standard protocols. For each experiment, 1 μM of RK-701, 0.5 mg protein/mL of microsome, and 3.5 μM β-NADPH were used, and the remaining substrate after 30 min of incubation was determined by LC–MS/MS analysis. [Dog/Monkey] Liver microsome analyses (dog, monkey) were performed in duplicate by RIKEN according to the same protocols as SCAS.

## Crystal structure determination

For crystal structure analysis, the human G9a SET domain (residues 913–1193) was synthesized using the *E. coli* cell-free synthesis method[48,49] and purified in the same manner as described previously[50], except that the final size-exclusion column chromatography using HiLoad Superdex 200 16/60 (GE Healthcare) was equilibrated with 20 mM Tris-HCl buffer (pH 8.0) containing 400 mM NaCl. The purified G9a SET domain (300 μM) was mixed with sinefungin (the cofactor analog) and RK-701 in a molar ratio of 1:5:5 for several hours before the crystallization setup. Crystals were grown by the hanging drop method at 293 K, where the sample solution was mixed with an equal volume of a reservoir solution containing 100 mM Bis-Tris propane (pH 7.4), 200 mM sodium formate, 10% ethylene glycol, and 25% PEG 3350. The harvested crystals were coated with Paratone-N oil (Hampton Research) before being flash-cooled in liquid nitrogen. Diffraction data were collected at the X06DA beamline of the Swiss Light Source, Paul Scherrer Institut (Villigen, Switzerland), and were processed and scaled with XDS[51], AIMLESS[52], and other CCP4 programs. The structure of G9a with UNC0638 (PDB ID: 3RJW)[23] was used for the initial phase determination by the molecular replacement method with Phaser[53]. The topology and parameter file for the inhibitor was generated with eLBOW[54] from SMILES strings. Structure refinement was carried out with Phenix[55] and manual model building using COOT[56]. Crystallographic statistics are summarized in Supplementary Table 3. The coordinates and structure factors of the G9a–RK-701 complex have been deposited in the PDB under accession code 7X73.

## Surface plasmon resonance (SPR) measurement

SPR was performed using a Biacore T200 (Cytiva) and Series S Sensorchip SA (Cytiva, BR100531) at the temperature of 25 °C. A solution of PBS, 0.005% TWEEN20 (Sigma, P9416), 0.5 mM TCEP (Nacalai Tesque, 06342-21), and 2% DMSO (Sigma, 276855) was filtered with a 0.22-μm filter and used as the running buffer. Kinetic measurements were carried out at a flow rate of 30 μL/min, and the association and dissociation phases were monitored for 120 s and 120–300 s, respectively. The G9a SET domain was biotinylated by using NHS-PEG4-Biotin (Thermo, 39259), and captured to flow cells 2, 3, and 4 with immobilization levels of ~2300 RU. No specific treatment was applied to the reference surface (flow cell 1). To prevent oxidation of the immobilized protein, a solution containing 1 μM $ZnCl_2$ and 5 mM TCEP was injected at a flow rate of 10 μL/min for 60 s before each set of kinetic measurements. The extra wash command was executed after each measurement cycle with 50% DMSO aqueous solution. To correct for bulk responses, solvent correction was performed once after all kinetics runs by using running buffer containing 1–3% DMSO. The sensorgrams obtained from the three flow cells were analyzed individually by Biacore T200 Evaluation Software (Cytiva). The dissociation constant ($K_D$) and binding kinetic parameters were calculated by a curve fit to a built-in 1:1 binding model.

## Cell viability assay

A WST-8 (2-(2-methoxy-4-nitrophenyl)-3-(4-nitrophenyl)-5-(2,4-disulfophenyl)-2H-tetrazolium) assay using a Cell Counting Kit-8 (DOJINDO) was performed to measure cell viability using an EnSpire microplate reader (PerkinElmer).

## Animal experiments

All animal experiments were conducted in accordance with the animal experimental protocol and guidelines of the Tokyo University of Pharmacy and Life Sciences Animal Experimentation Regulations after review by the Institutional Animal Care and Use Committee

(permission numbers L18-10 and L19-26) and approval by the President of the Tokyo University of Pharmacy and Life Sciences.

C57BL/6 mice (6–8 weeks old, female, six mice/group) were purchased from Tokyo Laboratory Animals Science. The mice were fed a standard laboratory diet and housed in a temperature-controlled room under specific pathogen-free conditions. A combination anesthetic was prepared with 0.3 mg/kg medetomidine, 4.0 mg/kg midazolam, and 5.0 mg/kg butorphanol.

Mice were injected intraperitoneally with 0, 20, or 50 mg/kg RK-701 for 5 consecutive days, followed by 2 days off. The dosing solution contained 15% DMSO, 17.5% Cremophor EL, 8.75% EtOH, 8.75% HCO-40, and 50% PBS. Blood samples were collected once a week by Retro Orbital Sampling (75 µL Calibrated Micropipette; Drummond Scientific Company) during isoflurane anesthesia. Blood RNA was extracted using NucleoSpin RNA Blood (Takara Bio Inc.). The weight of each mouse was measured every 3 or 4 days.

## Pharmacokinetic (PK) assay

PK studies were performed using 8-week-old male Crl:CD1 ICR mice in accordance with the animal experimental protocol, and procedures were approved by the Animal Care and Ethics Committee of Sekisui Medical. RK-701 was dosed to three mice at 25 mg/kg by intraperitoneal injection, as previously reported[57]. Blood samples were collected at 1, 2, 4, 6, and 24 h following administration and prepared into plasma by centrifugation. RK-701 in samples were analyzed by liquid chromatography-tandem mass spectrometry (LC–MS/MS) after deproteinization with acetonitrile. Plasma concentrations of RK-701 were determined using an internal standard (IS) method with a calibration curve of 1–30,000 ng/mL and niflumic acid as IS. The intra-day accuracy and precision of RK-701 were performed by analyzing three replicates of quality control samples (3, 300 and 10,000 ng/mL). PK parameters were calculated using the program Phenix WinNolin (Certara).

The LC-MS/MS system comprised of ACQUITY Ultra Performance LC (Waters) and 4000 QTRAP (Sciex). Acquity UPLC BEH C18 (2.1 × 50 mm, 1.7 µm, Waters) was used as analytical column. The mobile phase consisted of 0.1% formic acid in water (mobile phase A) and acetonitrile (mobile phase B) at a total flow rate of 0.5 mL/min with a column temperature of 40 °C. The chromatographic separation was achieved using gradient elution. The gradient program was: 20% B from 0.0–0.1 min, 20–95% B from 0.1–1.5 min, 95–95% B from 1.5–2.0 min, and 20% B from 2.01–3.0 min. The MS/MS instrument was operated in positive-ion mode with an ion spray voltage of 4200 V, entrance potential of 10 V and source temperature at 600 °C. The curtain gas was set at 12 psi, ion source gas 1 at 40 psi, ion source gas 2 at 50 psi. The respective optimum MS parameters for each analyte were as follows: declustering potential of 76 and 81 V, collision energies of 41 and 31 V, and collision cell exit potential of 12 and 14 V for RK-701 and IS, respectively. The multiple reaction monitoring transitions monitored for RK-701 were $m/z$ 447.2 > 204.2 and IS 283.1 > 265.1. Data acquisition and quantitation were performed by using Analyst (ver.1.4.3, Sciex).

## Mouse acute toxicity assay

All animal studies were approved by the Institutional Committee for Animal Experiments of the Institute of Microbial Chemistry, and were performed in accordance with relevant guidelines and regulations to minimize animal suffering. Four-week-old female ICR mice were purchased from Charles River Breeding Laboratories (Yokohama, Japan) and maintained in a specific pathogen-free barrier facility. Mice were maintained in a pathogen-free environment (23 ± 2 °C, 55 ± 5%) on an 11-h light/13-h dark cycle with food and water supplied ad libitum throughout the experimental period. Samples were dissolved in saline containing 10% DMSO and 1% Tween 80 and injected intravenously into mice.

## Ames test

The Salmonella mutagenicity assay was performed by pre-incubating the test compounds for 20 min with *Salmonella typhimurium* strains TA98 and TA100, with and without metabolic activation (S9 mixture). RK-701 was tested at the following concentrations: 4.88; 19.5; 78.1; 313; 1250; 2500; and 5000 µg per plate.

The various concentrations of RK-701 (100 µL) to be tested were added to either 500 µL of 0.1 M phosphate buffer (pH7.4) with 100 µL of bacterial culture and then incubated at 37 °C for 20 min. After this time 2 mL of top agar was added to the mixture and poured on to a plate containing minimum agar. The plates were incubated at 37 °C for 49 h and the His ± revertant colonies were counted manually and automatically by a colony counter (Colony analyzer CA-11D systems, System science Co.,Ltd.). The influence of metabolic activation was tested by adding 500 µL of 10% S9 mixture. All experiments were performed in duplicate. The standard mutagens used as positive controls in experiments without S9 mix were 2-(2-Furyl)-3-(5-nitro-2-furyl)acrylamide (0.01 µg/plate) for TA100 and (0.1 µg/plate) for TA98, and benzo[a]pyrene (5 µg/plate) with TA100 and TA98 in the experiments with metabolic activation. DMSO served as the negative control (100 µL/plate). The mutagenic index (MI, the average number of revertants per plate divided by the average number of revertants per plate from the negative control) was also calculated for each dose. A sample was considered positive when the MI was equal to or greater than 2 for at least one of the tested doses.

## In vivo micronucleus assay

Bone marrow cells treated with RK-701 were obtained from the right femur of 5 male rats of 2 weeks repeated oral administration study, and doses were evaluated at 10, 30 and 100 mg/kg via gavage administered to animals. A negative control group was established, where the animals received orally 5% methylcellulose solution. After 24 h of the last administration of RK-701, the bone marrows were collected, and smear preparation fixed with methanol was prepared. We counted erythrocytes, recorded the frequencies of polychromatic erythrocyte (PCE) cells in 500 erythrocytes and micronucleated polychromatic erythrocyte (MNPCE) cells in 4000 PCE cells, and calculated the average frequency and the standard deviations for each treatment groups.

## Construction of plasmids

PrimeSTAR Max DNA polymerase (Takara) was used for PCR. The RNA was extracted from HUDEP-2 cells, and cDNA was prepared using ReverTra Ace (TOYOBO). The BGLT3 gene was amplified with cDNA as a template and subcloned into the EcoRI/NotI sites of the retroviral expression vector pCX4-pur. BCL11A and ZBTB7A genes were subcloned into the EcoRI/XbaI sites of the pIRES-3xFLAG, and CHD3 and CHD4 genes were subcloned into the EcoRI/XhoI sites of the pcDNA3.1(-)/myc-his B. The retroviral vector pSUPER (OligoEngine) was used for stable expression of shRNA. RNAi plasmids targeting G9a (5′-CTACTCTGTGGATGAGCGC-3′) and BGLT3 (5′-CATCGATCTCTGCTCTTAA-3′), and non-targeting (5′-AAATCACAGAATCGTCGTA-3′) shRNA as a negative control, were obtained with annealed oligos and ligation in accordance with the manufacturer's instructions. The sequences of the oligonucleotides for gene are described in Supplementary Table 11. The oligonucleotides were purchased from INTEGRATED DNA TECHNOLOGIES or FASMAC.

## Retroviral infection

Retroviral infection was used as previously described to establish cell lines stably expressing BGLT3 and transfected with the pCX4-pur vector, as well as knockdown cell lines transfected with the pSUPER vector[58]. Briefly, retroviruses were produced by transfecting PLAT-E cells with pCX4-pur-BGLT3 or pSUPER-ctrl, pSUPER-BGLT3, or pSUPER-G9a using 5 µg/mL polyethylenimine MAX (PEI MAX,

Polysciences). Two days after transfection, the culture supernatant was collected and filtered through a 0.45-μm filter (Millipore) and used for infection with 8 μg/mL polybrene (Sigma). Forty-eight hours after infection, the infected cells were selected with 0.2 μg/mL puromycin (InvivoGen). The clonal puromycin-resistant cells were established via limiting dilution. The desired clones were confirmed by western blotting or qPCR.

## Protein identification and quantification by LC–MS/MS

The extraction and digestion of proteins were performed according to the method reported in a previous paper[59]. Briefly, collected cell pellets, 120 μL of water, 12.5 μL of acetonitrile, and 12.5 μL of 1% formic acid were mixed and sonicated. The mixture was centrifuged to remove the cell debris and the supernatant was collected. Ten microliters of supernatant was reduced by dithiothreitol, alkylated with iodoacetamide, and digested with trypsin/Lys-C mix (Promega) in 50 mM ammonium bicarbonate (pH 8.0) (Sigma) at 37 °C overnight. For protein quantification analysis by LC–MS/MS, 10 pmol of stable isotope–labeled peptides (synthesized by the Support Unit for Bio-Material Analysis, RIKEN Center for Brain Science) were added to the digested sample as an internal standard. The digested sample was stored at −25 °C until analysis. The trypsinized sample was applied to an EASY-nLC 1000 liquid chromatograph (Thermo Fisher Scientific) and Q-Exactive mass spectrometer (Thermo Fisher Scientific) equipped with a nanospray ion source (Thermo Fisher Scientific). The peptides were separated with a NANO–HPLC Capillary Column C18 (0.075 mm i.d. × 150 mm, 3 μm; Nikkyo Technos) using a 120 min gradient at a flow rate of 300 nL/min: 5–35% B in 100 min, and then 35–65% B in 20 min (solvent A, 0.1% formic acid; solvent B, acetonitrile/0.1% formic acid). The resultant MS and MS/MS data were searched against the Swiss-Prot database (2020_01) using the MASCOT software (Matrix Science). The search parameters were as follows: enzyme, trypsin; max. missed cleavages, 1; fixed modifications, carbamidomethyl (Cys); variable modifications, oxidation (Met); peptide mass tolerance, 6 ppm; fragment mass tolerance, 20 mmu; min. peptide length, 4. The proteins were considered to be identified when they have at least one peptide with a significant MASCOT score ($p < 0.05$).

Protein quantification was performed using Vanquish UHPLC (Thermo Fisher Scientific) and TSQ-Vantage EMR (Thermo Fisher Scientific) with a HESI-II ion source (Thermo Fisher Scientific) in multiple reaction monitoring (MRM) mode. MRM transitions for target peptides and internal standards are summarized in Supplementary Table 12. The peptides were separated with a YMC-Triart C18 column (2.0 mm × 50 mm, 1.9 μm; YMC) at 40 °C using a linear gradient at a flow rate of 0.4 mL/min: initially 2% B for 1 min, and then 2–37% B for 4 min (solvent A, 0.1% formic acid; solvent B, acetonitrile). Pinpoint software (Thermo Fisher Scientific) was used for peak area calculation.

## Flow cytometry

For the analysis of HbF expression, HUDEP-2 cells or cells differentiated from human CD34 + progenitor cells were fixed in 0.05% glutaraldehyde for 10 min, then washed two times with PBS containing 2% FBS (PBS/2% FBS) and permeabilized with 0.1% Triton X-100 (prepared in PBS/2% FBS; Life Technologies) for 5 min. After washing with PBS/2% FBS two times, cells were stained with allophycocyanin (APC)-conjugated HbF antibody (1:1, Invitrogen, MHFH05). One million cells were incubated with the HbF–APC antibody for 30 min in the dark at 4 °C. Cells were washed twice with PBS/2% FBS, then suspended in 0.2 ml PBS/2% FBS and analyzed using a FACSCalibur (Becton Dickinson). To assess erythroid differentiation, hematopoietic cells differentiated from human CD34 cells were stained with FITC-conjugated CD45 antibody (1:1, Becton Dickinson, 555482) and APC-conjugated glycophorin A antibody (1:20, Becton Dickinson, 551336). Stained cells were suspended in 0.2 ml PBS/2% FBS containing 1 μg/ml propidium iodide (Sigma) and analyzed by

FACSCalibur. The results of flow cytometer were further analyzed by FlowJo software (Becton Dickinson).

## Chromatin immunoprecipitation (ChIP) assay

A total of $0.5–1 \times 10^6$ cells were cross-linked with 1% formaldehyde for 10 min at room temperature and the reaction was quenched with 250 mM glycine. Cross-linked cells were then lysed with buffer 1 (15 mM Tris-HCl (pH 7.5), 60 mM KCl, 15 mM NaCl, 5 mM MgCl₂, 0.1 mM EGTA, 300 mM sucrose, 500 mM DTT, 1 mM phenylmethanesulfonyl fluoride (PMSF), 0.5% NP40) and incubated on ice for 10 min. After centrifugation, the pellet was re-suspended in buffer 2 (15 mM Tris-HCl (pH 7.5), 60 mM KCl, 15 mM NaCl, 5 mM MgCl₂, 0.1 mM EGTA, 1.2 M sucrose, 500 mM DTT, 1 mM PMSF). After centrifugation, the pellet was then re-suspended in MNase buffer (50 mM Tris-HCl (pH 7.5), 1 mM CaCl₂, 4 mM MgCl₂, 320 mM sucrose, 1 mM PMSF) and digested with 100 U of micrococcal nuclease (Takara) at 37 °C for 15 min. After the reaction was quenched with 20 mM EDTA, the supernatants were incubated with antibodies (anti-H3K9me2 (1:100), Abcam, ab1220; anti-H3K9ac (1:100), Active Motif, 39917; Goat anti-IgG (H + L) (1:100), Jackson Immunoresearch; anti-BCL11A (1:100), Novus Biologicals, NB600-261; anti-ZBTB7A (1:100), Thermo Fisher Scientific, eBioscience (13E9); anti-CHD3 (1:100), Novus Biologicals, NB-100-60412; anti-CHD4 (1:100), abcam, ab240640) and Protein A/G PLUS-Agarose (Santa Cruz Biotechnology) overnight at 4 °C. Beads were washed sequentially once with the following buffers: wash buffer A (50 mM Tris-HCl (pH 7.5), 75 mM NaCl, 10 mM EDTA, 0.01% NP40), wash buffer B (50 mM Tris-HCl (pH 7.5), 100 mM NaCl, 10 mM EDTA, 0.01% NP40), wash buffer C (50 mM Tris-HCl (pH 7.5), 175 mM NaCl, 10 mM EDTA, 0.01% NP40). After treatment with proteinase K in buffer 4 (20 mM Tris-HCl (pH 8.0), 10 mM EDTA, 400 mM NaCl, 0.5% SDS) at 56 °C overnight, DNA was purified using the FastGene Gel/PCR Extraction Kit (Nippon Genetics). The samples were analyzed by qPCR. Primers are listed in Supplementary Table 10. ChIP-seq experiments were performed by GENEWIZ. Tracks were visualized using the integrative genome viewer. ChIP-seq analysis were analyzed Cutadapt (V1.9.1) for QC, FastQC (V0.10.1) for QC assessment, Bowtie (V2.1.0) for mapping, strand cross-correlation (SCC) for ChIP quality assessment, MACS2 for peak calling, and R package ChIPpeakAnno for Peak annotated.

## Immunoprecipitation (IP)

HEK293T cells transfected with plasmids were lysed in NETN buffer (50 mM Tris-HCl (pH 7.5), 150 mM NaCl, 5 mM EDTA, 0.1% NP-0.1) and the cell lysates were incubated with anti-FLAG M2 affinity gel (1:50, Sigma, A2220) overnight at 4 °C. The beads were then washed with NETN buffer, and the bound proteins were extracted with SDS-PAGE loading buffer by heating at 95 °C for 5 min.

## In vitro pull-down assay

HUDEP-2 cells were lysed in NETN buffer. Pre-equilibrated streptavidin sepharose beads (Cytiva) were incubated with 3 μg of each biotinylated histone peptide for 30 min at 4 °C in NETN buffer containing 0.5% BSA (Sigma). The cell lysates were added, and the reaction mixture was incubated overnight at 4 °C. After beads were washed with lysis buffer, bound proteins were extracted with the SDS-PAGE loading buffer by heating at 95°C for 5 min. Proteins were separated by SDS-PAGE followed by immunoblot analysis. Biotinylated histone H3 peptide (1–21) was purchased from Sigma. Biotinylated histone H3K9me2 peptide (ARTKQTARK(me2)STGGKAPRKQLAGGK-biotin) was synthesized by the Support Unit for Bio-Material Analysis in RIKEN Center for Brain Science, Research Resources Division (RRD).

## RNA-seq analysis

Total RNAs were extracted using an RNeasy Plus Mini Kit (QIAGEN). RNA-seq experiments were performed by GENEWIZ or Rhelixa. Gene Ontology (GO) analysis in Supplementary Fig. 7a, the statistical test

was performed by Wallenius in goseq software. In Supplementary Tables 8(1-3), the statistical test (adjp) was determined using multiple comparisons analysis, Benjamini-Hochberg adjusted FDR test of the default of AltAnalyze (Ver. 2.0).

### Sequence alignment and phylogeny analysis

Sequences of the β-globin locus region in 12 mammal species were obtained from the National Center for Biotechnology Information (NCBI) database. The DNA sequences were aligned using the default options of ClustalW. The phylogenetic tree was drawn based on the neighbor-joining (NJ) method (1000 bootstrap repeats). Multiple sequence alignment was assessed by JalView[60]. Phylogenetic analysis was visualized using Molecular Evolutionary Genetics Analysis software. The timing of β-like globin expression (embryonic and fetal) and the hierarchies of biologically relevant groupings are based on multiple reports in the literature[61–63].

### Statistical analysis

Error bars indicate mean ± standard deviation (SD) unless otherwise mentioned. Significance was determined using one-way analysis of variance (ANOVA) with Tukey's post hoc test, two-tailed t-test, or Williams' test.

### Compound synthesis

A detailed description of the synthetic procedures of the compounds is included in the Supplementary information.

### Reporting summary

Further information on research design is available in the Nature Portfolio Reporting Summary linked to this article.

## Data availability

The data that support this study are available from the corresponding authors upon reasonable request. The raw RNA-seq and ChIP-seq reads obtained in this study have been submitted to the DDBJ Sequence Read Archive (DRA) under accession number DRA014673. The coordinates and structure factors of the G9a–RK-701 complex have been deposited in the PDB under accession code 7X73. Superimposing the structures of G9a–RK-701 and G9a–UNC0638 (PDB ID: 3RJW). The mass spectrometry proteomics data have been deposited to the ProteomeXchange Consortium via the PRIDE[64] partner repository with the dataset identifier PXD033536. Source data are provided with this paper.

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

## Acknowledgements

We thank the staff at the X06DA beamline of the Swiss Light Source, Paul Scherrer Institute (Proposal No. 20172015), for their help with X-ray diffraction data collection, and Dr. Hiroo Koyama (Drug Discovery Chemistry Platform Unit, RIKEN CSRS), Dr. Yoshinobu Hashizume, Dr. Toshio Goto, and Dr. Hiroshi Okazaki (RIKEN Program for Drug Discovery and Medical Technology Platforms) for their expert support and advice. We are grateful to the Support Unit for Bio-Material Analysis, RIKEN CBS Research Resources Division, for technical help with peptide synthesis. We thank Dr. T. Kitamura of The University of Tokyo for providing the PLAT-E cells, and Dr. Maeda of Kyusyu University for providing the BCL11A and ZBTB7A KO HUDEP-2 cells. This work was carried out as a research program under the RIKEN Program for Drug Discovery and Medical Technology Platforms and the Project for Development of Innovative Research. This work was supported by Platform for Drug Discovery, Informatics, and Structural Life Science from the Ministry of Education, Culture, Sports, Science and Technology, Japan; the Project for Development of Innovative Research on Cancer Therapeutics (P-DIRECT) from AMED to A.I.; the Pioneering project ("Epigenome manipulation") from RIKEN to Y.S. and A.I.; a Grant-in-Aid for Scientific Research (S) (19H05640) to M.Y. and a Grant-in-Aid for Young Scientists (JP20K15451) to S.T. from the Japan Society for the Promotion of Science (JSPS); and a Grant-in-Aid for JSPS Research Fellow (JP21J01114) to S.T.

## Author contributions

S.T. performed most of the experiments and wrote the initial manuscript. T.H. performed the FACS analysis and helped write the manuscript. Y.M., A.N., M.A., and T. So. provided biochemical data. T. So. performed HTS. F.S., Y. Ni., T. Su., R.N., and N.H. designed and synthesized chemical compounds and helped write the manuscript. S.M. performed the SPR analysis and helped write the manuscript. H.N., S.S., T.U., and M.S. performed the crystal structure analysis and helped write the manuscript. F.S. and M.U. performed the MS analysis and helped write the manuscript. T.O., S.O., and M.K. performed the mouse acute toxicity study and helped write the manuscript. Y.H. and H.H. provided technical support for animal studies. T.Y. and Y.S. supervised the experiments involving RK-701. Y. Na. helped design experiments and supervised the experiments involving CD34+ cells. M.Y. conceived and designed the

study and edited the manuscript. A.I. conceived and designed the study and wrote the manuscript.

## Competing interests

F.S., Y. Ni., T.S., N.H., R.N., M.Y., and A.I. are inventors on a PCT international patent application (WO/2021/106988) that covers RK-701, a compound studied in this paper. The remaining authors declare no competing interests.
