## [Peer Review File · Nature Communications]

REVIEWER COMMENTS

Reviewer #1 (Remarks to the Author):

In this manuscript, the authors developed a highly selective and nongenotoxic G9a inhibitor, RK-701, which induced fetal globin expression both in human erythroid cells and in mice. They confirmed that BGLT3 lncRNA (already known) has an essential role in γ -globin induction under the treatment of RK-701, hydroxyurea and other chemicals. They also showed that RK-701 upregulated BGLT3 by inhibiting the recruitment of two known γ -globin repressors, BCL11A and ZBTB7A, onto the BGLT3 gene locus.

The experiments are well executed and logically analyzed, which make a contribution to understanding the mechanisms of the γ -globin repression and derepression. However, the biological importance of BGLT3, BCL11A/ZBTB7A in γ -globin regulation has been reported previously. The therapeutic potentiality and usefulness of RK-701 in the treatment of sickle cell anemia model remains unsolved. Deep molecular insight and the biological significance of G9a/GLP inhibitor will be required.

Major points:

1. Although the authors mentioned that RK-701 is a specific and low-toxicity G9a inhibitor, there is a lack of detail demonstration for the screening, identification and the characterization of chemical properties. Because the development of this compound is the most central and novel point in this study, the authors should accurately state chemical biology aspects, rather than the application and reevaluation part regarding the known mechanism of γ -globin gene locus.

2. There are emerging queries in the core part: fundamental information of RK-701 (reversible or irreversible inhibition, duration of action on G9a/GLP, effect on G9a/GLP protein levels (Fig 1c), etc.); possible reason for a low-toxicity of RK-701 compared with UNC0638; side effects in RK-701-treated mice due to non-selective genome-wide action (Fig 2f); separate effects of RK-701 on transcriptional/epigenomic changes and erythroid differentiation (Ext Fig 4c); similarity and difference between RK-701 and G9a-KD experiments. Thus, the authors should present such basic and chemical data of this inhibitor.

3. In Fig 3, the authors presented RNA-seq data and should perform a GSEA pathway analysis, which can reveal whether BGLT3 and HBG genes were significant or not (Fig 3a, Table S2). There are many genes with higher fold changes or q values. Transcription and epigenetic factors can be regulated at protein levels, unlike BGLT3 lncRNA. To verify the inhibitory effect of RK-701, the authors can perform an RNA-seq of G9a-KD HUDEP-2 cells, BCL11A KO and ZBTB7A KO cells (Fig 3d, Fig 4f), and compare the overlapping or non-overlapping genes on genome.

4. This reviewer felt the major concern about reevaluation of BCL11A and ZBTB7A in globin locus control. Since the contribution of these factors to globin expression was already demonstrated, this part would be not-so-new. Especially, the functional relationships between G9a/GLP, H3K9me2 and these transcription factors remain unclear. What is the recruitment mechanism of BCL11A and ZBTB7A to the target sites in BGLT3 gene? How did RK-701 act on these mechanisms? Therefore, the authors need to conclude the action mechanism of RK-701.

Reviewer #2 (Remarks to the Author):

Manuscript by Takase et al "A novel and specific G9a inhibitor unveils BGLT3 lncRNA as a universal mediator of chemically induced fetal globin gene expression" describes the discovery of a new G9a/GLP inhibitor and its use in fetal globin reactivation to treat sickle cell disease.

RK-701 characterization is extensive, although authors do not provide the replicate numbers for the in-vitro potency or western blot experiments. Perhaps the high Kd obtained in SPR experiments could be explained better.

Where possible, the authors should provide experimental replicates for western blot and flow cytometry experiments.

In Fig 3a volcano plot, some of the data points seem to be out of scale.

The authors decided to focus their efforts on the known regulators of fetal hemoglobin regulators BGLT3, BCL11A and ZBTB7A and provide convincing evidence to support that these regulators are responsible for the RK-701 effect on fetal hemoglobin.

Several questions remain that are raised in the discussion. Given that BCL11A and ZBTB7A directly bind to DNA how would H3K9me2 affect this binding? Perhaps as the authors allude, chromatin accessibility using should be investigated. ATAC-seq could be used for this. More discussion of possible binding partners and possible involvement of DNA methylation would also be interesting.

Finally, is there cooperativity between BCL11A and ZBTB7A? Would knockdown of one affect the binding of the other?

Overall the manuscript is well written, presents a novel compound and applies it in an interesting area, thus I fully support the publication.

Reviewer #3 (Remarks to the Author):

In this manuscript, Takase et al, show the development of a new inhibitor against G9a and demonstrate its regulation in the role played by the lncRNA BGLT3 in the induction of γ -globin. The work carried out is interesting but needs to be completed with additional experiments to better understand and assess the importance of the results obtained.

- The authors show practically all of their results in a single cell line, HUDEP-2, and some of the experiments in CD34 + cells. In order to better understand the potential of these results, all these experiments should be performed on at least 2-3 different cell lines.

- The authors show in Figure 1c that their inhibitor RK-701 produces an inhibition of H3K9me2, the G9a-dependent histone modification. But have the authors analyzed other histone modifications such as H3K27me3? Although the inhibitor RK-701 does not inhibit EZH2 biochemically, several studies have shown that G9a and EZH2 can form a complex and therefore the inhibition of G9a could also lead to a decrease in H3K27me3.

- The authors to compare their results obtained with their G9a inhibitor RK-701, use the UNC0638 as commercially available G9a inhibitor. But, what about other G9a inhibitors such as CM-272 that not show cytotoxic effects? There are currently several inhibitors against G9a and it would be necessary to include some other inhibitors in the comparative study. In relation to the toxicity studies, have the authors studied whether RK-701 produces any type of hematological or hepatic toxicity? Has it been analyzed whether the inhibition of G9a produced by RK-701 is reversible or not? This would be very interesting to know if unwanted effects could occur in a long treatment, such as 5 weeks.

- How do the authors explain that Hydroxyurea increases H3K9me2 levels and also HbF levels?

- How do the authors explain the effect that the increased binding of ZBTB7A in the HS3 and HS2 regions of the β -globin locus could have after treatment with RK-701? These regions do not affect the expression of the γ -globin lncRNA? How do the authors explain the decreased binding of BCL11A and ZBTB7A after treatment with RK-701? What is the mechanism? What is the direct role of a G9a inhibitor in this process?

- Due to the fact that the expression of the BGLT3 increases after treatment with different epigenetic inhibitors, it would be very interesting to analyze in greater depth the epigenetic regulation suffered by the β -globin locus, and specifically of the BGLT3 region, in order to be able to define better the inhibitors that could have the best effect in increasing the expression of BGLT3 and γ -globin.

Reviewer #4 (Remarks to the Author):

The authors discovered a novel, potent, and selective G9a inhibitor with favorable pharmacokinetic profile. They used it as a chemical tool to demonstrate that G9a inhibition upregulates BGLT3 by inhibiting the recruitment of γ -globin repressors to the BGLT3 genomic loci, which in turn induces fetal γ -globin, with potential therapeutic applications against Sickle cell disease.

While reactivation of fetal γ -globin by G9a inhibitors is not new, the compound and extensive mechanistic characterizations are, which, in my opinion, deserves publication in this Journal.

Major comment:

While binding of the compound to G9a in a novel binding pose is clearly established crystallographically, the KD measured by SPR is not convincing, and not in line with the IC50 value measured in the methyltransferase assay. The SPR data is noisy and RU values are unacceptably low. Maybe not enough protein was loaded on the chip. This experiment should be repeated, maximum expected RU values provided and KD re-evaluated.

Reviewer #1:

Major points:

1. *Although the authors mentioned that RK-701 is a specific and low-toxicity G9a inhibitor, there is a lack of detail demonstration for the screening, identification and the characterization of chemical properties. Because the development of this compound is the most central and novel point in this study, the authors should accurately state chemical biology aspects, rather than the application and reevaluation part regarding the known mechanism of γ -globin gene locus.*

According to the reviewer's suggestion, we have added details for HTS and chemical properties of RK-701 including solubility, permeability, and metabolic stability. These results have been presented in the new Supplementary Figure 1, Supplementary Table 4, and Supplementary Table 9. In addition, we have added the result of SAR analysis, which explains the early optimization process that led to generating RK-701. The result has been shown in the new Supplementary Table 1. Because we performed optimization studies by synthesizing a considerable number of derivatives (more than 1,000 compounds) and subsequent huge SAR analyses, it is obviously difficult to include all the data in this manuscript and we plan to submit details of the optimization process to another journal related to medicinal chemistry. In fact, we have already prepared the manuscript and it is ready for submission as attached.

2. *There are emerging queries in the core part: fundamental information of RK-701 (reversible or irreversible inhibition, duration of action on G9a/GLP, effect on G9a/GLP protein levels (Fig 1c), etc.); possible reason for a low-toxicity of RK-701 compared with UNC0638; side effects in RK-701-treated mice due to non-selective genome-wide action (Fig 2f); separate effects of RK-701 on transcriptional/epigenomic changes and erythroid differentiation (Ext Fig 4c); similarity and difference between RK-701 and G9a-KD experiments. Thus, the authors should present such basic and chemical data of this inhibitor.*

Regarding reversibility, SPR analysis clearly indicates that RK-701 is a reversible inhibitor. This result has been presented in the new Supplementary Figure S2a. We have stated it in the text of the revised manuscript.

For the duration of action on G9a/GLP, we calculated the K_{off} value of RK-701 on G9a from the results of SPR analysis. This result has been shown in the new Supplementary Figure S2a of the revised manuscript.

Regarding G9a/GLP protein levels, we tested the effect of RK-701 on protein levels of G9a and GLP. The result demonstrated that RK-701 did not affect their protein levels. This result has been shown in Figure 1f of the revised manuscript.

To further investigate fundamental information about RK-701, we have also performed kinetic analyses, showing that RK-701 acts as a histone H3 substrate-competitive inhibitor. This result has been presented in the new Supplementary Figure 3 of the revised manuscript.

The reason for the low toxicity of RK-701 compared with UNC0638 is currently unknown. However, it seems possible that due to the difference in the chemical structure, RK-701 and its metabolites have lower off-target effects than other inhibitors. Please see Complementary Figure 1 at the bottom of this letter. UNC0638, UNC0642, A-366, and CM-272 are 7-(3-(pyrrolidin-1-yl)propoxy)quinazoline derivatives that share a similar framework structure. These compounds exhibited strong toxicities as shown in Supplementary Figure S3a-c, suggesting that their common substructure is the cause of toxicities. We are currently planning to conduct GLP preclinical studies including animal toxicity tests with long-term administration.

Regarding the side effects of RK-701 in the treated mice, we checked body weight (Fig. 2h) and the appearance of the general conditions of mice treated with RK-701 for 5 weeks, but no significant abnormalities were observed. In addition, we conducted a 2-week repeated dose oral toxicity study on rats and did not find any obvious abnormality including hematological and hepatic parameters. Thus, we have not detected any side effects of RK-701 so far. These results of hematological and hepatic parameters of rats treated with RK-701 for 2 weeks have been shown in the new Supplementary Table 5 of the revised manuscript.

Regarding the separate effects of RK-701 on transcriptional/epigenomic changes and erythroid differentiation, we have checked the effect of RK-701 on erythroid cell differentiation from CD34⁺ cells on day 11 in addition to days 14 and 17. RK-701 did not affect erythroid differentiation on any day, unlike the globin-inducing effects. These results on erythroid differentiation on days 11, 14, and 17 have been presented in the new Supplementary Figure 5h of the revised manuscript.

In response to the reviewer's comment on "similarity and difference between RK-701 and G9a-KD experiments", we performed RNA-seq analysis in G9a-knockdown HUDEP-2 cells and compared it with gene expression changes in RK-701-treated

HUDEP-2 cells. Both γ -globin and BGLT3 were commonly upregulated in RK-701-treated and G9a knockdown cells. These results have been shown in the new Supplementary Figure 7b and Supplementary Tables 7 and 8 of the revised manuscript.

3. In Fig 3, the authors presented RNA-seq data and should perform a GSEA pathway analysis, which can reveal whether BGLT3 and HBG genes were significant or not (Fig 3a, Table S2). There are many genes with higher fold changes or q values.

Transcription and epigenetic factors can be regulated at protein levels, unlike BGLT3 lncRNA. To verify the inhibitory effect of RK-701, the authors can perform an RNA-seq of G9a-KD HUDEP-2 cells, BCL11A KO and ZBTB7A KO cells (Fig 3d, Fig 4f), and compare the overlapping or non-overlapping genes on genome.

Thank you for the constructive comments. According to the reviewer's suggestion, we performed GO analysis on RNA-seq data from RK-701-treated cells. This analysis revealed that hemoglobin complex among GO terms was significantly enriched in RK-701-treated cells. This may reflect that RK-701 induces both two fetal and embryonic globin genes. On the other hand, "(positive and negative) regulation of hemoglobin biosynthetic process" of GO term was not enriched in this analysis. Thus, GO analysis also suggests selective induction of BGLT3 among globin regulators. These results have been shown in Supplementary Figure 7a of the revised manuscript.

We also performed RNA-seq analyses in BCL11A KO and ZBTB7A KO cells and compared them with gene expression changes in RK-701-treated HUDEP-2 cells. These results have been presented in the new Supplementary Figure 7b and Supplementary Table 8 of the revised manuscript.

4. This reviewer felt the major concern about reevaluation of BCL11A and ZBTB7A in globin locus control. Since the contribution of these factors to globin expression was already demonstrated, this part would be not-so-new. Especially, the functional relationships between G9a/GLP, H3K9me2 and these transcription factors remain unclear. What is the recruitment mechanism of BCL11A and ZBTB7A to the target sites in BGLT3 gene? How did RK-701 act on these mechanisms? Therefore, the authors need to conclude the action mechanism of RK-701.

Thank you for your fundamentally important question. In response to the Reviewer's

comments, we carried out multiple approaches to elucidate the mechanism underlying the BGLT3 gene repression by BCL11A and ZBTB7A and the reactivation of BGLT3 expression by G9a inhibition, and found two critical results. First, the H3K9me2 level at the BGLT3 gene locus was selectively and significantly reduced in BCL11A KO or ZBTB7A KO cells. Indeed, we found that both BCL11A and ZBTB7A bind to G9a in cells thereby facilitating G9a-induced H3K9me2 at the BGLT3 gene locus. Second, CHD4, a component of the NuRD complex that is known to recognize histone H3K9 methylation, was highly occupied on the BGLT3 gene locus. Importantly, RK-701 reduced not only histone H3K9 methylation, but also CHD4 occupancy at the BGLT3 gene locus. Together with the results that CHD4 can bind to both BCL11A and ZBTB7A, we concluded that CHD4 mediates, at least in part, the recruitment of BCL11A and ZBTB7A to the BGLT3 gene locus. However, an H3K9me2-independent CHD4 recruitment mechanism may also exist because RK-701 only partially reduced CHD4 occupancy at the BGLT3 gene locus. These results have been presented in new Figure 4b-d and Figure 5c-f, and described in the results and discussion sections of the revised manuscript. This positive feedback loop mechanism that the H3K9me2-dependent binding of BCL11A and ZBTB7A via CHD4 further recruits G9a onto the BGLT3 gene locus may result in the enhancement of locus-specific H3K9me2 and establishment of the repressive chromatin state.

Based on these results altogether, we have proposed a new model of the molecular mechanism by which G9a represses fetal globin gene expression as shown in new Figure 8. We believe that these findings strengthen our manuscript and greatly appreciate the reviewer's comments that lead to the discovery of the new molecular mechanism of fetal globin repression by G9a.

Reviewer #2 (Remarks to the Author):

Manuscript by Takase et al “A novel and specific G9a inhibitor unveils BGLT3 lncRNA as a universal mediator of chemically induced fetal globin gene expression” describes the discovery of a new G9a/GLP inhibitor and its use in fetal globin reactivation to treat sickle cell disease.

RK-701 characterization is extensive, although authors do not provide the replicate numbers for the in-vitro potency or western blot experiments. Perhaps the high Kd obtained in SPR experiments could be explained better.

Where possible, the authors should provide experimental replicates for western blot and flow cytometry experiments.

According to the reviewer's suggestion, we have indicated the replicate number of all data in the figure legends. We have also provided experimental replicates for western blot and flow cytometry experiments as resource data and have shown a representative result in the figures.

As for SPR, we carefully re-analyzed with modified experimental conditions in response to reviewer #4's suggestion and got data indicating a lower Kd value (0.75 μ M) than before (2.4 μ M), although it is still higher than the IC50 value (27 nM). The reason for the difference between IC50 and Kd values is currently unknown. However, the presence of SAM may affect these values since kinetic analyses suggest that RK-701 inhibits G9a uncompetitively with respect to SAM (Please see new Supplementary Figure 3b). Indeed, SAM was not included in the SPR experiments. We have discussed this point in the text and new SPR data has been shown in the new Supplementary Figure 2a of the revised manuscript.

In Fig 3a volcano plot, some of the data points seem to be out of scale.

Some of the data points as indicated by the reviewer are infinity which is out of scale because gene expression levels at the steady state are almost 0. This explanation has been added to the figure 3 legend of the revised manuscript.

The authors decided to focus their efforts on the known regulators of fetal hemoglobin regulators BGLT3, BCL11A and ZBTB7A and provide convincing evidence to support that these regulators are responsible for the RK-701 effect on fetal hemoglobin.

Several questions remain that are raised in the discussion. Given that BCL11A and ZBTB7A directly bind to DNA how would H3K9me2 affect this binding? Perhaps as the authors allude, chromatin accessibility using should be investigated. ATAC-seq could be used for this. More discussion of possible binding partners and possible involvement of DNA methylation would also be interesting.

We highly appreciate the reviewer's suggestion to us.

The reviewer's comment "Given that BCL11A and ZBTB7A directly bind to DNA how would H3K9me2 affect this binding?" prompted us to explore the possibility that BCL11A and ZBTB7A bring G9a to the BGLT3 region. As shown in the response to major point #4 from Reviewer #1, we found that both BCL11A and ZBTB7A bind to G9a and facilitate G9a-induced H3K9me2 at the BGLT3 gene locus and have proposed a new model of the molecular mechanism by which G9a represses fetal globin gene expression as shown in new Figure 8. In this new model, BCL11A and ZBTB7A recruit G9a and facilitate G9a-induced H3K9me2 at the BGLT3 gene locus. This G9a-induced H3K9me2 recruits BCL11A and ZBTB7A onto the BGLT3 gene locus via CHD4, which may further enhance locus-specific H3K9me2. This positive feedback loop of enhancement of H3K9me2 at the BGLT3 gene locus ensures the repression of both BGLT3 lncRNA and subsequent fetal globin expression. We believe that these findings strengthen our manuscript.

Regarding the possible involvement of DNA methylation, it is possible that DNA methylation is involved in RK-701-mediated HbF induction because a recent report suggests that G9a plays a role in maintaining DNA methylation (Cell Rep. 33, 108315, 2020). We appreciate the reviewer's beneficial comments and have discussed them in the text of the revised manuscript.

Finally, is there cooperativity between BCL11A and ZBTB7A? Would knockdown of one affect the binding of the other?

It has been reported that BCL11A and ZBTB7A regulate γ -globin expression independently (Science. 351, 285-289, 2016). Another report also suggests that the binding of one repressor does not affect the binding of the other (Nat. Genet. 50, 498–503, 2018). This explanation has been mentioned in the text of the revised manuscript.

Reviewer #3 (Remarks to the Author):

- The authors show practically all of their results in a single cell line, HUDEP-2, and some of the experiments in CD34 + cells. In order to better understand the potential of these results, all these experiments should be performed on at least 2-3 different cell lines.

To study the regulation of γ -globin gene expression using cell lines, we must select erythroid-like cells with low levels of γ -globin gene expression at the steady state. Among cell lines, HUDEP-2 is the only available one suitable for this purpose. We previously established several human erythroid progenitor cell lines derived from iPS cells (HiDEP-1 and 2) and CD34⁺ cells in umbilical blood (HUDEP1-3) (PLoS One. 8, e59890, 2013). Among them, only HUDEP-2 cells constitutively maintain low γ -globin gene expression (Fig. 7f). Thus, HUDEP-2 is the only commercially available erythroid progenitor cell line used worldwide to study the regulation of γ -globin gene expression. Indeed, most reports have used only HUDEP-2 cells in the experiments for γ -globin gene expression, which has been widely accepted by the hematology community. Although another cell line called BEL-A, generated from CD34⁺ hematopoietic progenitors, was most recently reported as an alternative cell line with constitutively low γ -globin expression (Cells 10, 523, 2021), it is, unfortunately, not commercially available at present. In this study, we used primary CD34⁺ cells and HiDEP-1, a human erythroid progenitor cell line derived from iPS cells with high γ -globin expression, to validate the results obtained from HUDEP-2. We agree with the reviewer's comment, but we hope that the reviewer could understand the reason why we used HUDEP-2 and its validity.

- The authors show in Figure 1c that their inhibitor RK-701 produces an inhibition of H3K9me2, the G9a-dependent histone modification. But have the authors analyzed other histone modifications such as H3K27me3? Although the inhibitor RK-701 does not inhibit EZH2 biochemically, several studies have shown that G9a and EZH2 can form a complex and therefore the inhibition of G9a could also lead to a decrease in H3K27me3.

According to the reviewer's suggestion, we checked the effect of RK-701 on H3K27me3. Please see the Complementary Figure 2 at the second panel from the

bottom of this letter. RK-701 seemed to slightly decrease H3K27me3 only at high concentrations. However, RK-701 had no effect on H3K27me3 at a concentration of 0.1 μ M, which was sufficient to reduce H3K9me2 and induce HbF (Fig. 2c and d). Therefore, a slight reduction, if any, of H3K27me3 by RK-701 is very likely to be independent of the HbF induction.

- The authors to compare their results obtained with their G9a inhibitor RK-701, use the UNC0638 as commercially available G9a inhibitor. But, what about other G9a inhibitors such as CM-272 that not show cytotoxic effects? There are currently several inhibitors against G9a and it would be necessary to include some other inhibitors in the comparative study. In relation to the toxicity studies, have the authors studied whether RK-701 produces any type of hematological or hepatic toxicity? Has it been analyzed whether the inhibition of G9a produced by RK-701 is reversible or not? This would be very interesting to know if unwanted effects could occur in a long treatment, such as 5 weeks.

According to the reviewer's suggestion, we tested the effect of CM-272 on cell toxicity, H3K9me2, and γ -globin and BGLT3 expression. Like RK-701, CM-272 induced both γ -globin and BGLT3 gene expression with a reduction of H3K9me2. In addition, we confirmed the reduction of H3K9me2 by UNC0638. However, CM-272, similar to UNC0638, exhibited severe cell toxicity against HUDEP-2 cells. These results have been depicted in the new Supplementary Fig. 4b and 8e-h of the revised manuscript. For hematological or hepatic toxicity, we checked the hematological or hepatic parameters of rats treated with RK-701 for two weeks. RK-701 did not affect all hematological or hepatic parameters tested. These results have been shown in the new Supplementary Table 5 of the revised manuscript.

Regarding reversibility, SPR analysis clearly indicates that RK-701 is a reversible inhibitor as shown in the new Supplementary Figure S2a. We have stated it in the text of the revised manuscript.

As for the unwanted effect of long treatment, we checked the body weight (Fig. 2h) and the appearance of the general condition of mice treated with RK-701 for 5 weeks, but no significant abnormalities were observed. In addition, a 2-week repeated dose oral toxicity study on rats was conducted, but no obvious abnormality including hematological and hepatic parameters mentioned above was observed. Thus, we have not detected any unwanted effects of RK-701 so far. We are currently planning to

conduct GLP preclinical studies including animal toxicity tests with long-term administration, which will be reported in a separate paper in the future.

- How do the authors explain that Hydroxyurea increases H3K9me2 levels and also HbF levels?

For evaluation of the effect of hydroxyurea on the H3K9me2 level, we quantified band intensities of H3K9me2 from independent three experiments. Please see the Complementary Figure 3 at the bottom of this letter. Hydroxyurea did not significantly affect H3K9me2 levels, while RK-701 significantly decreased them. These pictures from independent three experiments were submitted as resource data.

- How do the authors explain the effect that the increased binding of ZBTB7A in the HS3 and HS2 regions of the β -globin locus could have after treatment with RK-701?

As the reviewer mentioned, the binding of ZBTB7A in the HS3 and HS2 regions of the β -globin locus appears to be increased by RK-701, but these increases are not statistically significant, while the decrease in binding of ZBTB7A to the BGLT3 region is statistically significant. Therefore, we focused on the BGLT3 region in this manuscript.

These regions do not affect the expression of the γ -globin lncRNA?

We think that these regions, especially the HS2 region, may affect the expression of the BGLT3 lncRNA-mediated γ -globin gene. It is known that BCL11A binds to the HS2 region and suppresses γ -globin transcription by reconstituting the β -globin cluster through the regulation of the chromosomal loop formation. Because RK-701 drastically decreased the occupancy of BCL11A on HS2 in addition to the BGLT3 gene region (Fig. 5a), RK-701 may enhance the expression of the BGLT3 gene through a specific chromatin architecture induced by decreasing the binding of BCL11A to the HS2 region. This possibility has been discussed in the text.

How do the authors explain the decreased binding of BCL11A and ZBTB7A after treatment with RK-701? What is the mechanism? What is the direct role of a G9a inhibitor in this process?

Thank you for this important question, which was also pointed out by Reviewers #1 and #2. Therefore, please see the response to them. In brief, we found that both BCL11A and ZBTB7A recruit G9a and facilitate G9a-induced H3K9me2 at the BGLT3 gene locus. This G9a-induced H3K9me2 increases the binding of BCL11A and ZBTB7A to the BGLT3 gene locus via CHD4, which may further enhance locus-specific H3K9me2. This positive feedback loop of enhancement of H3K9me2 at the BGLT3 gene locus ensures the repression of both BGLT3 lncRNA and subsequent fetal globin expression. Inhibition of H3K9 methylation by RK-701 impairs the recruitment of the repressive complex containing BCL11A and ZBTB7A onto the chromatin of the BGLT3 gene locus as shown in the new Figure 8. We believe that these findings strengthen our manuscript.

- Due to the fact that the expression of the BGLT3 increases after treatment with different epigenetic inhibitors, it would be very interesting to analyze in greater depth the epigenetic regulation suffered by the β -globin locus, and specifically of the BGLT3 region, in order to be able to define better the inhibitors that could have the best effect in increasing the expression of BGLT3 and γ -globin.

Following the reviewer's suggestion, we tested the effect of SAHA on histone acetylation levels at the β -globin locus including the BGLT3 gene region. CHIP-qPCR analysis revealed that histone H3 acetylation at the β -globin gene locus including the BGLT3 gene region was increased by SAHA in HUDEP-2 cells. Thus, HDAC inhibitors induced BGLT3 gene expression by increasing histone acetylation at the BGLT3 gene locus. In addition, we tested a combination effect of RK-701 and decitabine on BGLT3 and γ -globin expression and found that treatment with both RK-701 and decitabine significantly increased the expression of both BGLT3 and γ -globin compared to each compound alone. These results suggest that combinations with different epigenetic inhibitors represent a useful strategy for the treatment of SCD. These results have been shown in the new Supplementary Fig. 11e-g of the revised manuscript.

Reviewer #4 (Remarks to the Author):

Major comment:

While binding of the compound to G9a in a novel binding pose is clearly established crystallographically, the KD measured by SPR is not convincing, and not in line with the IC50 value measured in the methyltransferase assay. The SPR data is noisy and RU values are unacceptably low. Maybe not enough protein was loaded on the chip. This experiment should be repeated, maximum expected RU values provided and KD re-evaluated.

We greatly appreciate the reviewer's thoughtful comment. We re-conducted an SPR analysis with a recently established more reliable assay system by using a streptavidin sensor chip and a biotinylated G9a. According to the reviewer's suggestion, measurements were performed up to the concentration at which the binding of G9a to the sensor chip was saturated. The RU value in this re-analysis is 7.7 ± 0.15 RU, while the theoretical RU value is 32 ± 0.13 RU. In catalog specification, however, Biacore T200 used in this study has a baseline noise root mean square of 0.03 RU. Therefore, we believe that the response levels of the obtained sensorgrams are sufficiently high. As the reviewer mentioned, increasing the protein immobilization level can lead to a higher theoretical Rmax, which in turn relatively reduces the noise and improve the sensorgram quality. On the other hand, it is known that the higher immobilization level may cause non-specific binding, mass transport limitation, and rebinding effect, which often decreases the reliability of kinetic parameters (Anal Biochem. 236, 275-283, 1996). Therefore, we believe that the re-evaluated Kd value is reliable. The new SPR data have been presented in the new Supplementary Figure 2a of the revised manuscript. The re-evaluated Kd value ($0.75 \mu\text{M}$) is lower than before ($2.4 \mu\text{M}$), but it is still higher than the IC50 value (27 nM). The reason for the difference between IC50 and Kd values is currently unknown. However, kinetic analyses suggest that RK-701 is uncompetitive with respect to SAM because of a decrease in IC50 values with increasing the SAM concentration (Please see the new Supplementary Fig. 4b). Therefore, its mode of uncompetitive inhibition with SAM may affect these values because IC50 and Kd values were measured in the presence and absence of SAM, respectively. We have added this explanation in the text of the revised manuscript.

Complementary Figure 1

Complementary Figure 3

Relative band intensity of H3K9me2 levels in CD34⁺ cells treated with RK-701 or hydroxyurea on day 10 of erythroid differentiation (related to Fig. 2e). Data are mean ± SD from three independent experiments. An asterisk indicates $p < 0.05$ determined using the one-way ANOVA with Tukey's post hoc test.

REVIEWERS' COMMENTS

Reviewer #1 (Remarks to the Author):

The authors had the best efforts on responding to the reviewers' comments. I think that the revised manuscript would be suitable for publication in the Journal.

Reviewer #2 (Remarks to the Author):

The concerns and suggestions have been addressed

Reviewer #4 (Remarks to the Author):

The authors have provided a valid interpretation for the relative weak binding of their compound observed by SPR: the binding event is measured in the absence of SAM. Indeed, IC50 values decrease with SAM, indicating a SAM-dependent binding mechanism. A similar structural mechanism was reported for other methyltransferases such as SETD7 (<https://pubmed.ncbi.nlm.nih.gov/25136132/>), PRMT5 (<https://pubmed.ncbi.nlm.nih.gov/25915199/>) or PRDM9 (<https://pubmed.ncbi.nlm.nih.gov/31848333/>), and reviewed in <https://pubmed.ncbi.nlm.nih.gov/31817960/>.

Once one of these references is cited, the manuscript is acceptable for publication in my opinion.

Response to Reviewer 4:

The authors have provided a valid interpretation for the relative weak binding of their compound observed by SPR: the binding event is measured in the absence of SAM. Indeed, IC50 values decrease with SAM, indicating a SAM-dependent binding mechanism. A similar structural mechanism was reported for other methyltransferases such as SETD7 (<https://pubmed.ncbi.nlm.nih.gov/25136132/>), PRMT5 (<https://pubmed.ncbi.nlm.nih.gov/25915199/>) or PRDM9 (<https://pubmed.ncbi.nlm.nih.gov/31848333/>), and reviewed in <https://pubmed.ncbi.nlm.nih.gov/31817960/>.

Thank you for sharing important reference information. According to the reviewer's suggestion, we have cited the above references in the text of the revised manuscript.